# Single-cell RNAseq analysis of spinal locomotor circuitry in larval zebrafish

**Jimmy J Kelly†, Hua Wen†, Paul Brehm***

Vollum Institute, Oregon Health & Science University, Portland, United States

**Abstract** Identification of the neuronal types that form the specialized circuits controlling distinct behaviors has benefited greatly from the simplicity offered by zebrafish. Electrophysiological studies have shown that in addition to connectivity, understanding of circuitry requires identification of functional specializations among individual circuit components, such as those that regulate levels of transmitter release and neuronal excitability. In this study, we use single-cell RNA sequencing (scRNAseq) to identify the molecular bases for functional distinctions between motoneuron types that are causal to their differential roles in swimming. The primary motoneuron, in particular, expresses high levels of a unique combination of voltage-dependent ion channel types and synaptic proteins termed functional 'cassettes.' The ion channel types are specialized for promoting high-frequency firing of action potentials and augmented transmitter release at the neuromuscular junction, both contributing to greater power generation. Our transcriptional profiling of spinal neurons further assigns expression of this cassette to specific interneuron types also involved in the central circuitry controlling high-speed swimming and escape behaviors. Our analysis highlights the utility of scRNAseq in functional characterization of neuronal circuitry, in addition to providing a gene expression resource for studying cell type diversity.

**\*For correspondence:**
brehmp@ohsu.edu

†These authors contributed equally to this work

**Competing interest:** The authors declare that no competing interests exist.

## eLife assessment

In zebrafish, primary motor neurons (PMNs) control escape movements, and a more heterogeneous population of secondary motor neurons (SMNs) regulate the speed of rhythmic swimming. Using single-cell RNA sequencing (scRNAseq), the authors have obtained **compelling** evidence that PMNs, and two types of interneurons innervating them, express a set of three genes encoding voltage-gated ion channels enabling rapid firing. The PMNs also express high transcript levels of proteins involved in exocytosis, which would be expected to support rapid neurotransmitter release. These results will be **important** for those working on spinal cord function and zebrafish genomics/transcriptomics.

## Introduction

Functional and anatomical studies of spinal circuitry among the vertebrates have formed the basis of our understanding of control over stereotypic movements (*Goulding, 2009*; *Grillner and Jessell, 2009*). Investigation into movement control continues to benefit from larval zebrafish, which represent a greatly simplified system for resolving the underlying spinal circuitry (*Fetcho and Liu, 1998*; *Drapeau et al., 2002*; *Lewis and Eisen, 2003*; *Fetcho et al., 2008*). Study of circuitry in zebrafish also provides the unique opportunity to trace locomotory circuitry from sensory initiation to the final motor output (*Fetcho, 1991*; *Koyama et al., 2011*; *Fidelin and Wyart, 2014*; *Berg et al., 2018*). Fortuitously, despite the evolutionary distance between fish and mammals, many classes of spinal interneurons involved in movement control are conserved between species, heightening the potential significance of zebrafish circuitry analysis (*Grillner, 2003*; *Goulding, 2009*).

As a new approach toward circuitry analysis, we turned to single-cell RNA sequencing (scRNAseq). For this purpose, we developed a method for isolation and dissociation of spinal cords from 4 days post fertilization (dpf) zebrafish, an age at which much of the anatomy and physiology of swim control has been published (*Bhatt et al., 2007*; *McLean et al., 2007*; *Fetcho et al., 2008*; *Liao and Fetcho, 2008*; *McLean et al., 2008*; *Satou et al., 2009*; *Menelaou and McLean, 2012*; *Wang and Brehm, 2017*; *Bello-Rojas et al., 2019*; *Menelaou and McLean, 2019*; *Kishore et al., 2020*; *Satou et al., 2020*; *Wen et al., 2020*). Our analysis has provided identification of major classes of neuronal and glial types. Of those neuronal types, many are known to be directly involved in locomotory circuitry either in mouse, zebrafish, or both. Using the markers revealed by the transcriptome analysis, we validated a number of key interneuron types, previously shown to be present in zebrafish locomotion circuit. In addition, we identified a new excitatory interneuron type that has a unique transmitter phenotype among interneurons.

Our interest in applying scRNAseq methodology to spinal neurons of zebrafish went beyond identifying transcriptional markers. Rather, we sought to use the neuron-specific transcriptomics to mine for distinctions in the ion channels and synaptic players that are associated with specialized circuit function. Many physiological studies have indicated fundamental differences in excitability and synaptic transmission among neurons controlling escape behavior versus those controlling slow to moderate rhythmic swimming speeds (*Bhatt et al., 2007*; *Liao and Fetcho, 2008*; *McLean et al., 2008*; *Satou et al., 2009*; *Menelaou and McLean, 2012*; *Menelaou and McLean, 2019*; *Kishore et al., 2020*; *Satou et al., 2020*). The signaling molecules causal to these differences among spinal neuron types are largely unknown. To address this outstanding question, we compared different types of motor neurons, where two well-studied subtypes serve different roles. The primary motoneurons (PMns) control the strongest contraction that provide for the single powerful bend, initiating escape, and participate in rhythmic swimming at only the highest speeds. By contrast, the secondary motoneurons (SMns) collectively regulate the range of slower rhythmic swimming (*McLean et al., 2007*; *Gabriel et al., 2011*; *Wang and Brehm, 2017*). Paired patch-clamp recordings, possible only in zebrafish, have characterized these Mn types as functional bookends. The PMns fire action potentials at ultrahigh frequency and the neuromuscular synaptic responses are able to follow with fidelity, whereas the SMns respond with lower frequency action potential firing and synaptic transmission at the neuromuscular junction and are subject to frequent failures (*Wang and Brehm, 2017*; *Wen et al., 2020*).

Our scRNAseq analysis, using Mn subtype-specific markers that we validate in this article, has provided candidates for serving these functional differences. First, the PMns express a trio of unique voltage-dependent ion channels, seen only at very low levels in the SMns, that are tailored for high-frequency transmission. The same fast ion channel cassette are also enriched in two well-characterized interneuron types that control firing of the PMns and are directly involved in the fast swimming and escape behavior. Second, the PMns also express significantly higher transcript levels of several key proteins involved in exocytosis, collectively termed a synaptic cassette. Thus, scRNAseq offers a new means to interrogate spinal circuitry through assignment of specialized signaling molecules. This application may also prove useful to understanding specialized circuits within the central nervous system (CNS) of mammals.

## Results

### Transcriptional profiling of the larval zebrafish spinal cord

scRNAseq was performed on 4 dpf larval zebrafish, an age corresponding to many studies of spinal circuitry and electrophysiological analysis of neuronal control over swimming behavior (*Bhatt et al., 2007*; *Liao and Fetcho, 2008*; *McLean et al., 2008*; *Satou et al., 2009*; *Menelaou and McLean, 2012*; *Wang and Brehm, 2017*; *Menelaou and McLean, 2019*; *Kishore et al., 2020*; *Satou et al., 2020*; *Wen et al., 2020*). For each of two duplicate experiments, ~150 spinal cords were isolated, followed by dissociation into single cells. In each case, ~10,000–15,000 spinal cord cells were sequenced to a depth of 40,000 reads per cell and aligned to an improved zebrafish reference genome that is more inclusive of 5′ and 3′ untranslated regions (*Lawson et al., 2020*; *Figure 1—figure supplement 1*; see 'Materials and methods'). After applying standard quality control filters and removing contamination from outside the spinal cord (*Figure 1—figure supplements 2 and 3*), data integration resulted in a total of 11,762 spinal cord cells from the combined datasets.

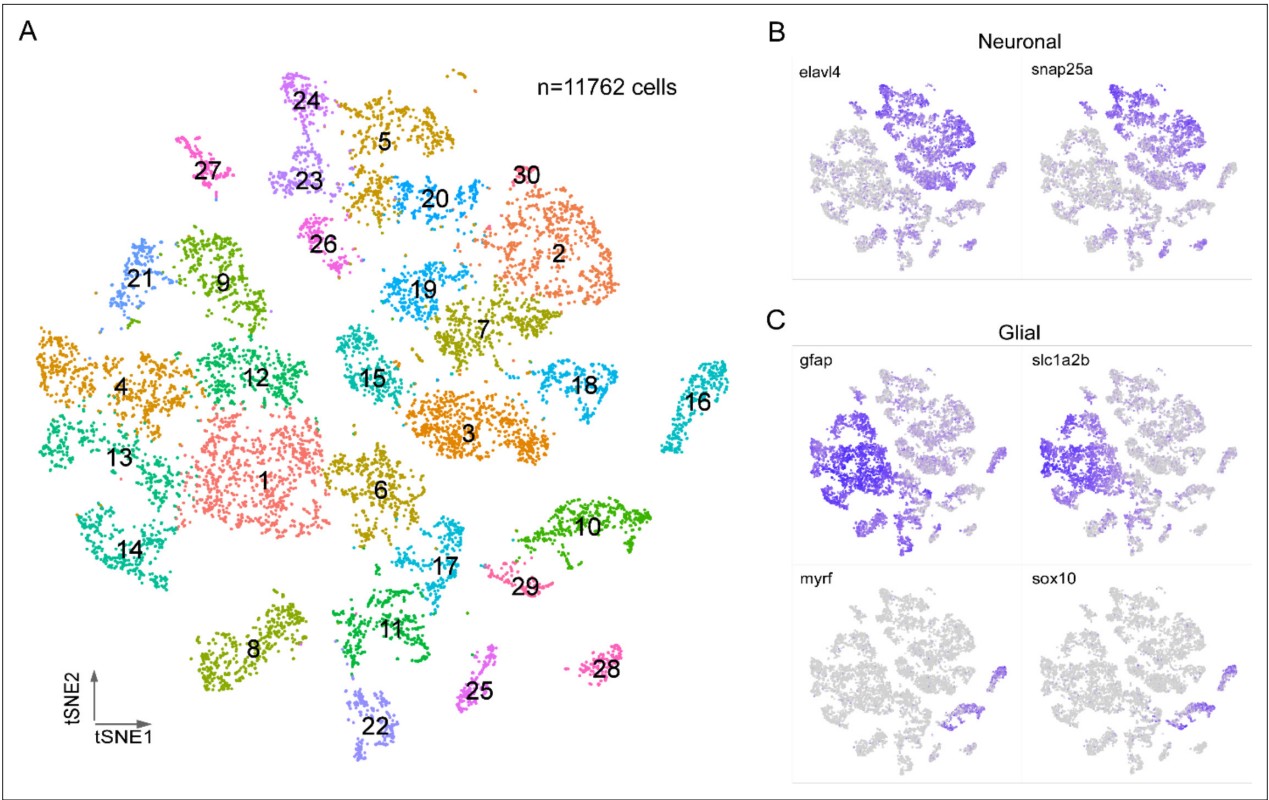

**Figure 1.** Transcriptional profiling of larval spinal cord. (**A**) Visualization of 4 days post fertilization (dpf) spinal cord cells using t-distributed stochastic neighbor embedding (t-SNE). Each dot is a cell and each arbitrary color corresponds to a single cluster. The clusters are individually numbered and the total number of cells indicated. (**B, C**) Feature plots for two neuron markers (**B**) and four glial makers (**C**). Two sets markers are shown to distinguish the two broad types of glial cells, *gfap* and *slc1a2b* for astrocytes/radial glia (**C**, top), *myrf* and *sox10* for oligodendrocytes (**C**, bottom).

The online version of this article includes the following figure supplement(s) for figure 1:

**Figure supplement 1.** Improved annotation of the *cacna1ab* gene to extend 3' untranslated region (UTR).

**Figure supplement 2.** Integration of datasets from duplicated experiments.

**Figure supplement 3.** Removal of non-spinal cells.

Graph-based, unsupervised clustering of the entire spinal cord transcriptome gave rise to 30 clusters, 27 of which were readily identifiable on the basis of established neuronal or glial markers forming the two broad categories of cell types (*Figure 1A*). The neuronal markers were *elavl4* and *snap25a* (*Figure 1B*) and glia markers were *gfap, slc1a2b, myrf,* and *sox10* (*Figure 1C*). The glial cells (46% of total cells) fell into two non-overlapping groups, corresponding to astrocytes/radial glia (*gfap+/ slc1a2b+),* and oligodendrocytes (*sox10+/myrf+*) (*Figure 1C*). Both glial types were composed of multiple clusters, indicating further diversity. The remaining three clusters (clusters 6, 11, and 17, corresponding to 11% of total cells) showed mixed expression of neuronal and glial markers. It is unclear whether this population reflects a true cell type or instead a group of unremoved doublets. Given that these clusters could not be assigned to either neurons or glia with confidence, they were excluded from the subsequent analysis. Additionally, while the glia data are available as a resource, they were not analyzed further in this study.

## Transcriptional profiling of spinal cord neurons

The neuronal population identified by *elavl4+*/*snap25a+* expression was regrouped into 33 clusters using Seurat (*Figure 2A*). To assign individual neuronal identities to these transcriptome clusters, we used a combination code that relied on the coexpression of neurotransmitter biosynthesis/transporter genes along with differentially expressed marker genes (DEGs; *Figure 2*, *Table 1*; *Supplementary file 1*). The first code provided clear separation of neuronal populations into four principal categories: glutamatergic (*slc17a6a+/slc17a6b+*), glycinergic (*slc6a5+*), GABAergic (*gad1b+/gad2+*),

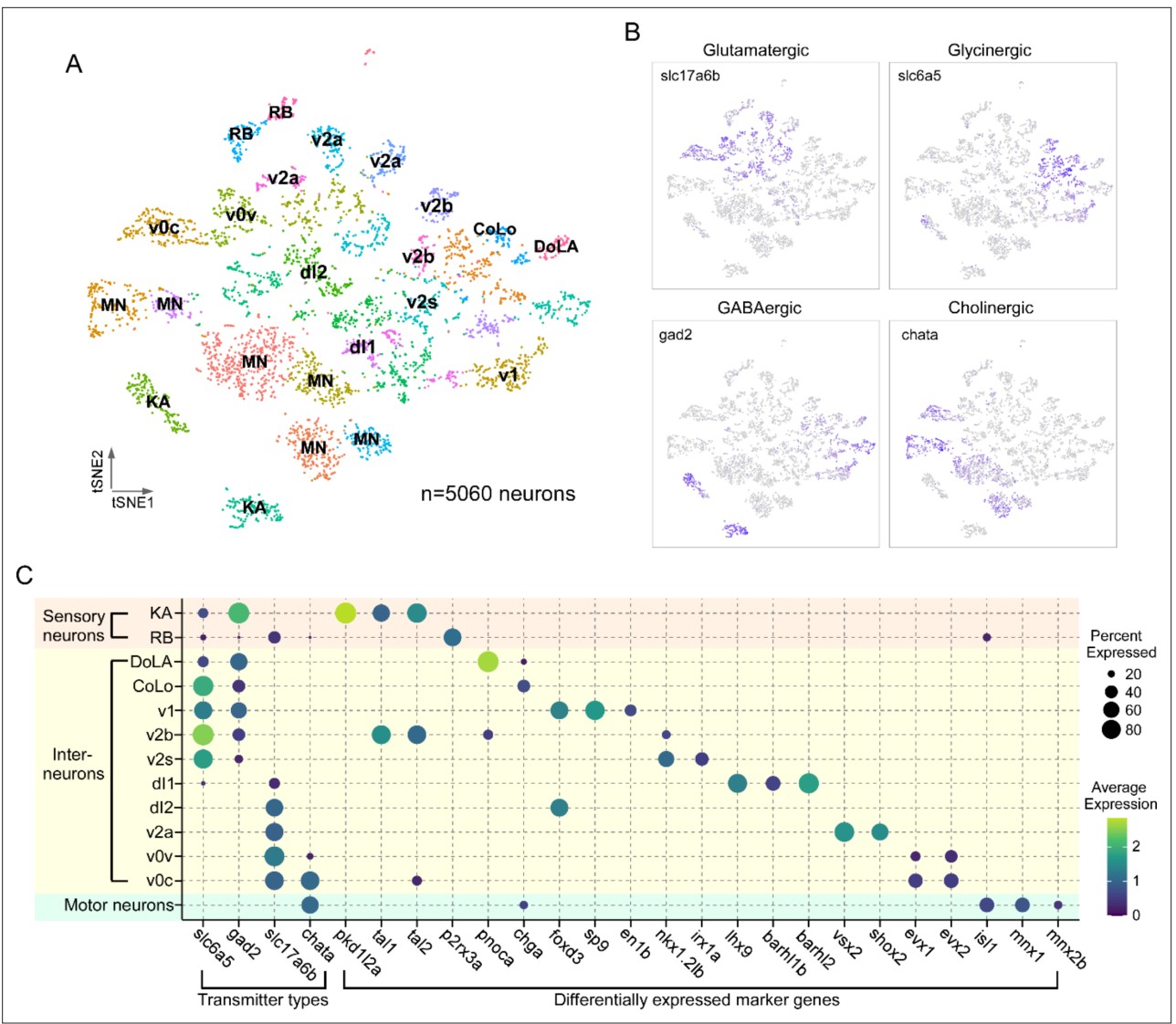

**Figure 2.** Transcriptional profiling of larval spinal cord neurons. (**A**) Visualization of neuronal populations for 4 days post fertilization (dpf) spinal cord using t-distributed stochastic neighbor embedding (t-SNE). Each dot is a cell and each arbitrary color represents a cluster. Cell type identity assigned to each cluster utilized the combination code of neurotransmitter phenotype, marker genes, and morphological labeling. (**B**) Feature plots for the four major classes of excitatory and inhibitory neurotransmitter genes. Vesicular glutamate transporter vGlut2 (*slc17a6b*) was used for glutamatergic neurons; glycine transporter glyt2 (*slc6a5*) for glycinergic neurons; glutamate decarboxylase (*gad2*) for GABAergic neurons; and choline acetyltransferase (*chata*) for cholinergic neurons. (**C**) Dot plot showing neuronal cell identity versus markers used for assignment. Dot size indicates the percentage of cells in the cluster showing expression of the indicated marker, and color scale denotes the average expression level. For visual clarity, dot sizes below 15% expressed are omitted.

or cholinergic (*chata+/slc18a3a+*) types (*Figure 2B*, *Table 1*). The second code relied on not only established markers for both zebrafish and mouse spinal neurons, but also new markers identified in this study. Generating the list of candidate markers was aided by previous studies that profiled the neurotransmitter identity of morphologically distinct neurons in the larval spinal cord (*Bernhardt et al., 1990*; *Hale et al., 2001*; *Higashijima et al., 2004a*; *Higashijima et al., 2004c*). In addition, since many aspects of the transcriptional program that establish the spinal neuronal circuit have been shown to be evolutionarily conserved among vertebrates (*Kiehn and Kullander, 2004*; *Goulding, 2009*; *Grillner and Jessell, 2009*), we cross-referenced our data with recent mouse spinal cord sequencing data to search for homologous marker genes (*Delile et al., 2019*; *Blum et al., 2021*).

Using the combination code, we identified specific classes of sensory neurons, Mns and inter-neurons (*Figure 2C*, *Table 1*). Two types of sensory neurons, the glutamatergic Rohon–Beard (RB)

**Table 1.** Combination codes used for assigning cell types to clusters.

| Cell type | Neurotransmitter | Markers | % of neurons |
| --- | --- | --- | --- |
| KA | GABA | pkd1l2a, tal1,tal2 | 7.0 |
| RB | Glutamate | p2rx3a | 3.8 |
| DoLA | GABA | pnoca | 1.4 |
| CoLo | Glycine | chga | 1.1 |
| v1 | Glycine/GABA | sp9, foxd3, en1b | 4.1 |
| v2b | Glycine | tal1, tal2 | 3.6 |
| v2s | Glycine | nkx1.2lb, irx1a | 2.7 |
| dl1 | Glutamate | lhx9, barhl1b, barhl2 | 1.7 |
| dl2 | Glutamate | foxd3 | 3.7 |
| v2a | Glutamate | vsx2, shox2 | 6.2 |
| v0v | Glutamate | evx1,evx2 | 4.0 |
| v0c | Glutamate/acetylcholine | evx1, evx2, | 4.2 |
| Mns | Acetylcholine | isl1, mnx1, mnx2b | 26.8 |

cells and GABAergic Kolmer–Agduhr (KA; also referred to as cerebrospinal fluid-contacting neurons [CSF-cN] cells), are each represented by two distinct clusters (*Figure 2A and C*). Cholinergic clusters formed a prominent group corresponding principally to Mns. Clusters of interneurons corresponded to the inhibitory v1, v2b, and v2s, and excitatory v0v, v2a, and dl1, and dl2 types (*Figure 2A and C*). The relative abundance of the cell types associated with individual clusters (*Table 1*) was consistent with those published for in vivo labeling experiments (*Higashijima et al., 2004b*; *Kimura et al., 2006*; *Ampatzis et al., 2014*; *Gerber et al., 2019*; *Satou et al., 2020*), suggesting that our preparation protocol and analysis are robust and unbiased in recovering spinal cell populations. Of the 33 clusters in our neuronal dataset, we were able to assign identities to 22 clusters with confidence. There are a number of cell types described for in larval zebrafish, such as the excitatory v3 interneuron or the inhibitory v0d interneuron (*Satou et al., 2020*; *Böhm et al., 2022*), that we were unable to identify likely due to lower expression levels of canonical markers at this developmental stage. It is probable that these interneuron populations are present in unassigned clusters.

Our analysis also resolved subtypes within neuronal classes. For example, the two clusters were assigned as KA neurons, which shared a common set of markers for the cerebrospinal fluid-contacting interneuron (*pkd1l2a/pkd2l1*; *Figure 3A*). However, assignment to subtype can be made on the basis

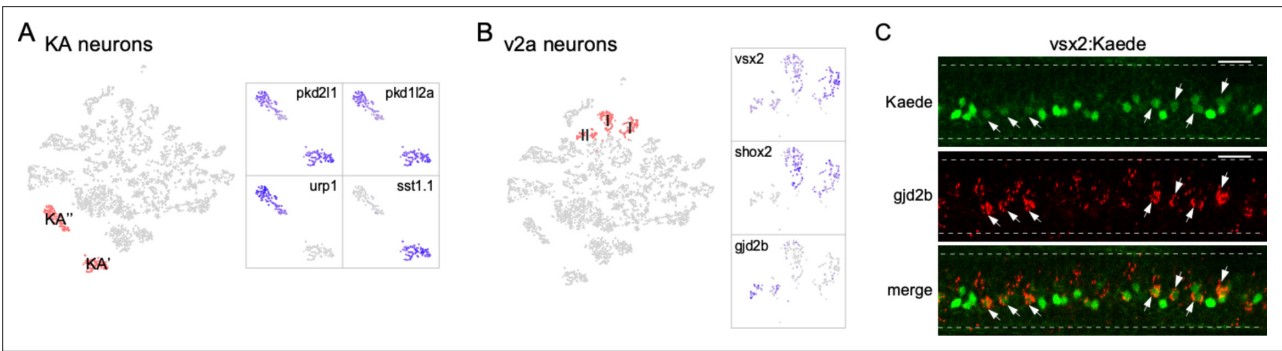

**Figure 3.** Diversity in neuronal types. (**A**) Zoomed feature plots for *pkd2l1, pkd1l2a, urp1, and sst1.1* that differentiate the KA' and KA'' neurons (right). The two clusters correspond to KA' and KA'' neurons indicated in the neuronal t-distributed stochastic neighbor embedding (t-SNE) projection (left, in red). (**B**) Zoomed feature plots for *vsx2, shox2, and gjd2b* that differentiate the type I and type II v2a neurons (right). The three clusters corresponding to v2a interneurons indicated in the neuronal t-SNE projection (left, in red). (**C**) Representative in situ hybridization images showing enriched expression of *gjd2b* in type II v2a (arrows) in a Tg(vsx2: Kaede) transgenic fish. The two subgroups of v2as were discerned with different levels green Kaede fluorescence. n = 8 fish. Scale bar 20 µm. Spinal cord boundary indicated with dashed lines.

of differential expression of *sst1.1* versus *urp1*, previously shown to label KA′ and KA″ functional groups, respectively (**Figure 3A**; **Djenoune et al., 2017**; **Yang et al., 2020**).

Similarly, the glutamatergic v2a interneurons consist of multiple clusters that represented distinct subtypes. This class of interneurons has previously been divided into two subpopulations, types I and II, based on morphology, molecular and functional heterogeneity (**Bhatt et al., 2007**; **McLean and Fetcho, 2009**; **Ampatzis et al., 2014**; **Menelaou et al., 2014**). One molecular feature differentiating the two types is the higher expression level of both the *vsx2* and *shox2* marker genes in type I compared to type II v2a (**Kimura et al., 2006**; **Menelaou et al., 2014**; **Hayashi et al., 2018**; **Menelaou and McLean, 2019**). In our dataset, both the *vsx2* and *shox2* expression patterns differed among the v2a clusters (**Figure 3B**). The two clusters with strong *vsx2* and *shox2* expression likely represent type I v2a neurons, while the third cluster with weak *vsx2* expression and no *shox2* expression likely represents type II (**Figure 3B**). For this cluster, we further identified *gjd2b*, the gene encoding the gap junction protein connexin 35.1δ subunit, as an additional marker gene (**Figure 3B**). This finding is consistent with previous immunohistochemistry in adult fish demonstrating selective expression of *gjd2* in type II v2a axons (**Carlisle and Ribera, 2014**; **Pallucchi et al., 2022**). Subtype assignment based on expression patterns of these markers was validated using in situ hybridization in larval Tg(vsx2: Kaede) fish (**Figure 3C**). Expression of the Kaede fluorescent protein driven by *vsx2* promoter labels type I v2a with strong fluorescence compared to weakly labeled type II (**Figure 3C**). Probes against *gjd2b* colocalized only with those neurons exhibiting weak fluorescence, confirming its specific expression in the type II v2a subtype (**Figure 3C**). The two subtypes of v2a interneurons both make direct connections with the PMns and are recruited during high-speed swimming but differ in terms of connection strength and the type II v2a interneuron is recruited more effectively across the range of high swimming speeds (**Kimura et al., 2006**; **Bhatt et al., 2007**; **McLean and Fetcho, 2009**; **Menelaou and McLean, 2019**). The availability of transcriptomes for these neuronal subgroups provides an opportunity to further mine for differential molecular features responsible for the functional distinction.

In addition to validating established markers for neuronal types and subtypes, our scRNAseq analysis revealed novel marker genes for identifying interneuron transcriptomes, as shown for the commissural local (CoLo) and dorsal longitudinal ascending (DoLA) interneurons.

## CoLo interneurons

CoLo interneurons provide the fast contralateral inhibition necessary for a successful escape response through direct contact with the PMns (**Fetcho and Faber, 1988**; **Liao and Fetcho, 2008**; **Satou et al., 2009**; **Kishore et al., 2020**). CoLo sends axons ventrally that turn to the contralateral side and split into short ascending and descending projections (**Higashijima et al., 2004a**; **Higashijima et al., 2004c**; **Liao and Fetcho, 2008**; **Satou et al., 2009**; **Kishore et al., 2020**; **Satou et al., 2020**; **Figure 4A1**). We found a small glycinergic cluster that expressed high levels of chromogranin A (*chga*) (**Figure 4A2**) that was distinct from the *chga*-enriched cholinergic cluster later assigned to the PMn type (**Figure 5D**). Labeling using *chga* in situ hybridization probes revealed a distinct large neuron located rostrally in each hemi-segment in addition to several large-sized motor neurons (**Figure 4A3**; see also **Figure 5D**). The stereotypical location and the one per hemi-segment stoichiometry of these *chga* labeled interneurons are consistent with that of CoLos (**Liao and Fetcho, 2008**; **Satou et al., 2009**; **Kishore et al., 2020**). To validate its identity, we transiently labeled inhibitory interneurons with EGFP under the control of the *dmrt3a* promoter (**Kishore et al., 2020**; **Satou et al., 2020**). Single GFP-labeled CoLos were screened based on morphology (**Figure 4A1**) and subsequently shown to co-label with *chga* (**Figure 4A3**).

## DoLA interneurons

DoLAs are GABAergic inhibitory interneurons located dorsally in spinal cord with well-described morphology (**Bernhardt et al., 1990**; **Higashijima et al., 2004c**; **Figure 4B1**). Our dataset indicated two GABAergic interneuron clusters, one of which corresponded to DoLA neurons with highly specific enrichment of *pnoca* expression (**Figure 4B2**). In situ hybridization using *pnoca* probes strongly labeled a small number of distinct dorsal neurons located immediately ventral to the domain occupied by the sensory RB cells (**Figure 4B3**). Sparse labeling of spinal neurons with an mCherry reporter showed that the neurons positive for *pnoca* projected long ascending axons with short ventral projections,

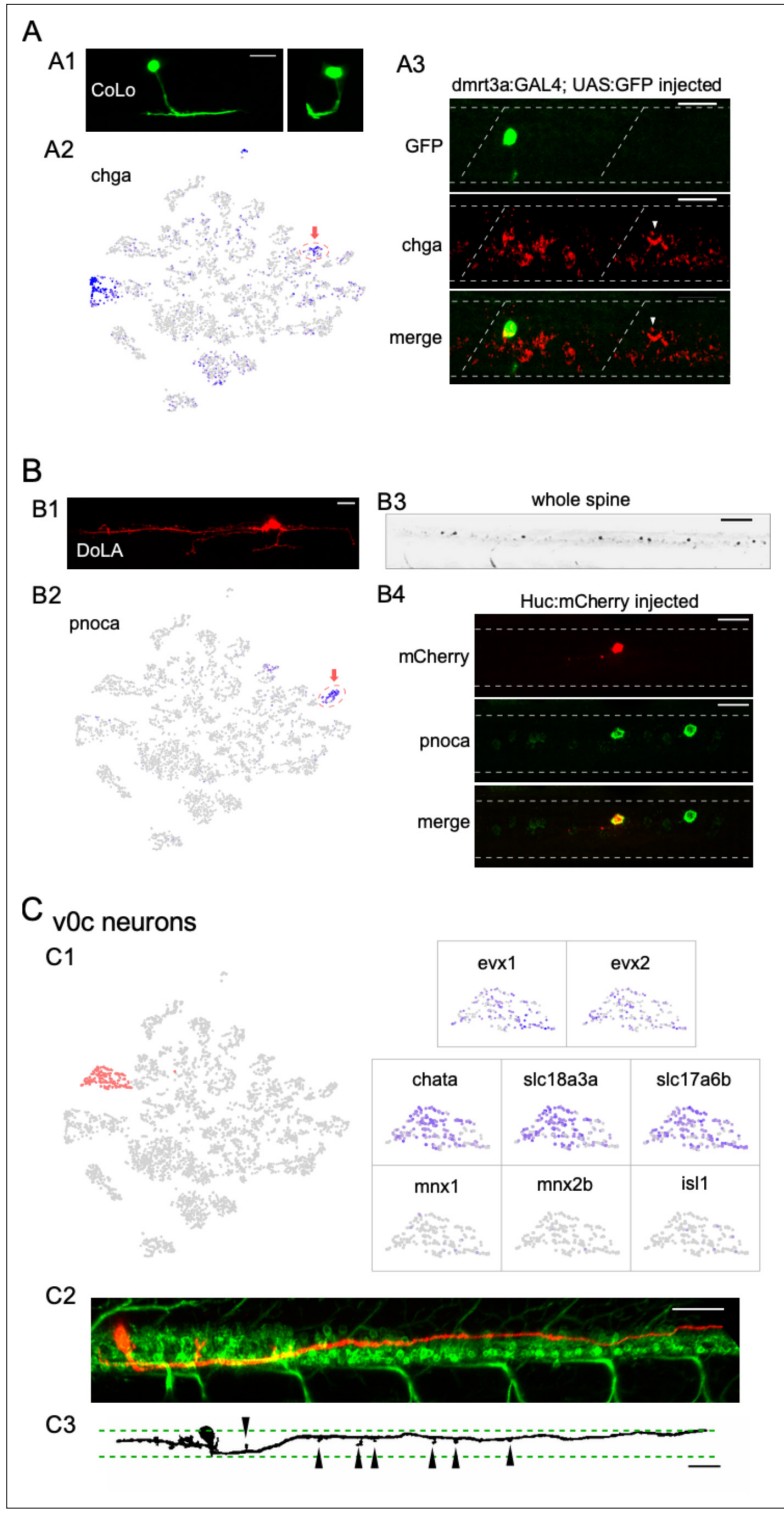

**Figure 4.** Identification of three different interneuron types using the combination code. (**A**) Commissural local (CoLo) interneurons. (**A1**) A CoLo neuron transiently labeled with GFP was identified by its short axons and localized commissural extension. A cross-section provides a clear view of its commissural branching (left). (**A2**) Feature plot of *chga* in the neuronal t-distributed stochastic neighbor embedding (t-SNE) projection. *chga* expression is localized in the CoLo cluster (red circle with arrow) in addition to a single motoneuron (Mn) cluster. (**A3**) *chga* in situ hybridization probes stained a CoLo labeled with GFP. The CoLo in the neighboring hemi-segment that was not labeled by GFP was also positive (arrowheads). Other positive labeling reflect the primary

*Figure 4 continued on next page*

*Figure 4 continued*

motoneurons (PMns) (see also *Figure 5D*). n = 6 fish. Boundary of spinal cord and segments indicated (white dash). Scale bar 20 µm. (**B**) Dorsal longitudinal ascending (DoLA) interneurons. (**B1**) A DoLA transiently labeled with mCherry was identified by its dorsal position and distinct morphology. (**B2**) Feature plot of *pnoca* in the neuronal t-SNE projection. *pnoca* expression is restricted in the DoLA cluster (red circle with arrow). (**B3**) In situ hybridization of *pnoca* shown for several spinal segments. n = 12 fish. Scale bar 100 µm. (**B4**) In situ hybridization of *pnoca* colocalized with an mCherry-labeled DoLA neuron. n = 7 cells. Scale bar 20 µm. (**C**) v0c interneurons. (**C1**) Zoomed feature plots for *evx1, evx2, chata, slc18a3a, slc17a6b, mnx1, mnx2b,* and *isl1* in the v0c cluster (right). The cluster corresponding to v0c neurons indicated in the neuronal t-SNE projection (left, in red). v0c interneuron cluster is identified by the coexpression of both glutamate (*slc17a6a*) and acetylcholine (*slc18a3a/chata*) pathway genes, and absence of Mn markers (*mnx1/mnx2b/isl1*). (**C2**) An example of a transiently labeled v0c by mCherry in a 4 day post fertilization (dpf) Tg(mnx1:GFP) fish. (**C3**) An example of v0c neurons in gray scale showing the morphology, with boundaries of the motor column (green dash) and enlargements along the axon (arrowheads) indicated. n = 37 fish. Scale bar 50 µm in (**C2**) and (**C3**). Caudal on right and rostral on left.

as well as with occasional short descending axons, thus matching the morphological characteristics reported for DoLA interneurons at this stage (*Figure 4B4*; *Bernhardt et al., 1990*; *Higashijima et al., 2004c*; *Wells et al., 2010*). The functional roles played by DoLA neurons have remained elusive. *pnoca* encodes Prepronociceptin, a precursor for several neuropeptides involved in multiple sensory signaling pathways (*Martin et al., 1998*). Its highly specific expression in DoLA suggests that they might function as neuropeptide-releasing neurons that modulate sensory functions in larval fish.

## v0c interneurons

Importantly, our analysis also identified a cholinergic/glutamatergic spinal interneuron type not described previously in larval zebrafish. A single cluster in our dataset expressed cholinergic markers that include vesicular acetylcholine transporter (vAChT) (slc18a3a) and choline acetyltransferase (chata), but lacks all canonical Mn markers (mnx1/mnx2b/isl1) (*Figure 4C1*). The cluster also expressed evx1 and evx2, markers associated with interneuron types in the v0 domain (*Figures 2C and 4C1*; *Zagoraiou et al., 2009*; *Juárez-Morales et al., 2016*). This transcriptional profile suggested that it represented the homologs to the premotor cholinergic v0c interneurons in mouse spinal cord (*Zagoraiou et al., 2009*). Cholinergic interneurons have only recently been shown to be present in adult fish by immunochemistry staining (*Bertuzzi and Ampatzis, 2018*). Similar to their mammalian counterparts, they play roles in modulating Mn excitability (*Bertuzzi and Ampatzis, 2018*). Notably, v0c in larval zebrafish differs from the mouse counterpart on the basis of coexpression of cholinergic and glutamatergic transmitter genes (e.g., slc17a6b, *Figure 4C1*).

We capitalized on the unique cholinergic phenotype of v0c among interneurons to provide in vivo labeling in the spinal cord. For this purpose, we injected a tdTomato reporter driven by the vAChT promoter to sparsely label cholinergic neurons in the 4 dpf spinal cord of Tg(mnx1: GFP) fish (*Figure 4C2*). In addition to the Mns, we observed mosaic labeling of an interneuron type with distinct position and morphology. The soma was located near the dorsal boundary of the motor column (*Figures 3 and 4C2*) and the axonal processes were either bifurcating (20 out of 37 cells) or purely descending (14 out of 37) or ascending (3 out of 37). The descending processes would reach lengths corresponding to multiple segments within the motor column (*Figure 4C2 and C3*, average length >6 segments). There was an overall lack of secondary branches, but enlargements reminiscent of synaptic boutons in close vicinity of Mn soma were observed along the neurites (*Figure 4C3*). Multiple neurons of this type were observed in the same segment even with the sparse labeling approach, suggesting that there are likely to be numerous v0cs in each segment. The morphology and anatomic arrangement are consistent with a role in modulating Mn properties, as has been proposed for v0c in both adult zebrafish (*Bertuzzi and Ampatzis, 2018*) and the mammalian homolog (*Zagoraiou et al., 2009*).

## Subclustering the Mns based on single-cell transcriptomes

We next focused on transcriptome comparison within Mn populations to examine their molecular heterogeneity. As a first approach, we isolated Mn clusters from the whole spinal cord dataset based on the overlap of two sets of marker genes (*Figure 5A*). The first set was based on components of the

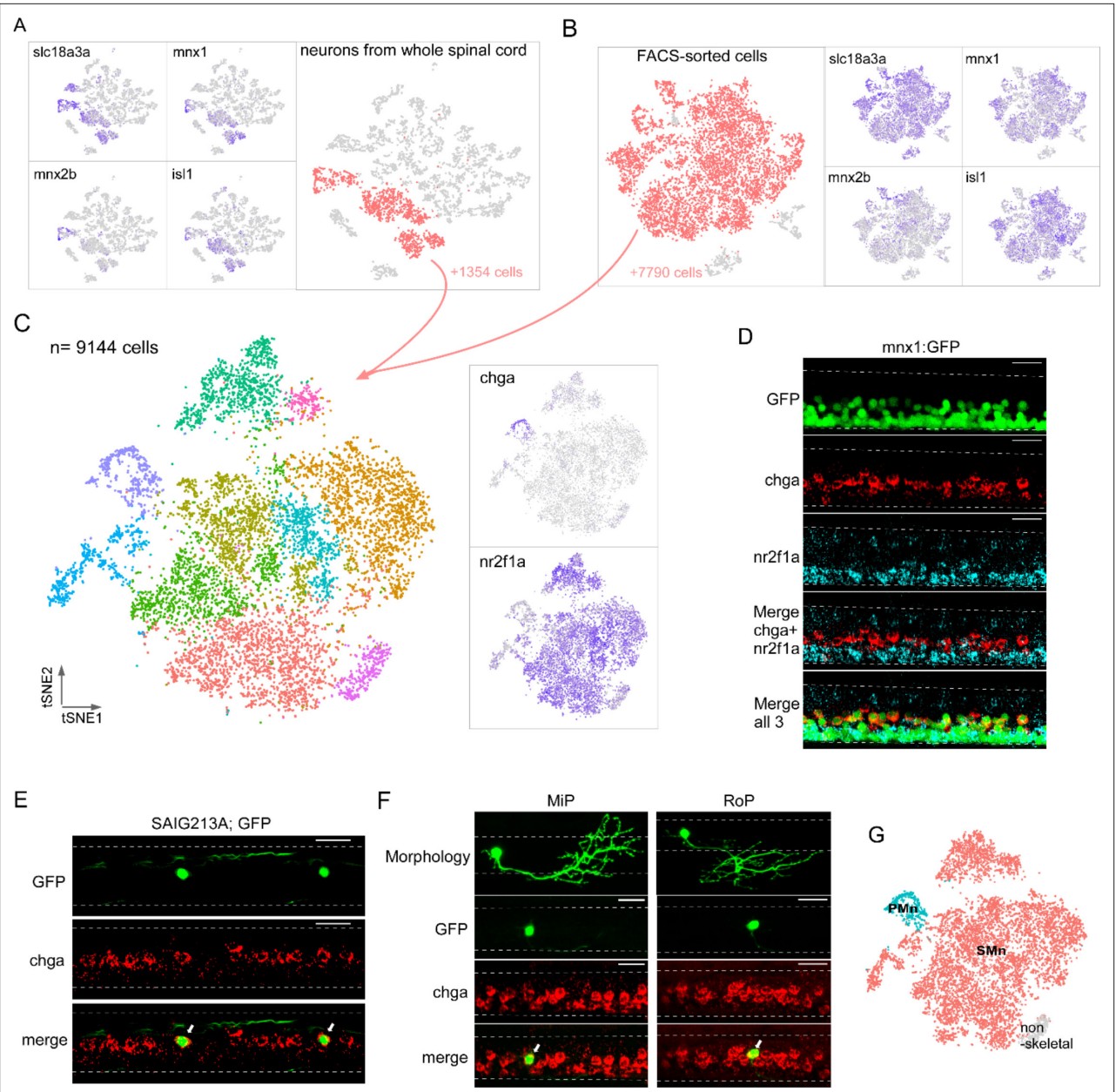

**Figure 5.** Single-cell transcriptional profiling of motoneuron (Mn) types at larval stage. Both computational extraction (**A**) and experimental enrichment (**B**) approaches were used to isolate Mn populations (red) on the bases of coexpression of acetylcholine transmitter genes (*slc18a3a* shown) and established Mn markers (*mnx1, mnx2b,* and *isl1*). The total numbers of Mns obtained using each approach indicated. (**C**) The integrated dataset shown in t-distributed stochastic neighbor embedding (t-SNE) projection, along with feature plots for two marker genes, *chga* and *nr2f1a*. (**D**) Representative in situ hybridization images using *chga* and *nr2f1a* probes in a 4 day post fertilization (dpf) Tg(mnx1:GFP) fish spinal cord. The motor column, indicated by GFP expression, is located ventrally in the spinal cord (top). *chga* and *nr2f1a* signals occupied more dorsal and ventral positions respectively within the motor column (bottom four panels). n = 13 fish. (**E, F**) in situ hybridization images showing specific expression of *chga* in primary motoneurons (PMns). Colocalization is shown for GFP-labeled CaP in Tg(SAIG213A;EGFP) fish (indicated by arrows in **E**, n = 14 fish), and individually labeled MiP and RoP (indicated by arrows in **F**, n = 4–6 cells). For images in (**D–F**), dorsal is up. Dashed line indicates the spinal cord boundary. Scale bar 20 µm. (**G**) PMn (cyan), secondary motoneurons (SMn) (red), and non-skeletal Mn (gray) assignment.

The online version of this article includes the following figure supplement(s) for figure 5:

**Figure supplement 1.** An unidentified cluster in the integrated motoneuron (Mn) datasets.

cholinergic pathway that included *slc18a3a* (*Figure 5A*) and *chata* (*Figure 2B*). The second set of Mn markers included the transcription factors *mnx1, mnx2b,* and *isl1* (*Figures 2C and 5A*; *Appel et al., 1995*; *Hutchinson and Eisen, 2006*; *Zelenchuk and Brusés, 2011*; *Asakawa et al., 2012*; *Seredick et al., 2012*). Overall, ~27% of the profiled neuronal single-cell transcriptomes corresponded to Mns (1354 cells) (*Figure 5A*). As a complementary approach, we also generated samples enriched for a larger number of Mns to increase the statistical power of the clustering analysis. This was achieved through fluorescence-activated cell sorting (FACS) of spinal cells prepared from the fluorescent trans-genic fish line, Tg(*mnx1:GFP*), that broadly labels Mns (*Flanagan-Steet et al., 2005*; *Bello-Rojas et al., 2019*). Clustering analysis of the sorted data showed that >92% of the FACS-sorted cells represent Mns based on the same canonical markers used for whole spinal cord, giving rise to 7790 cells for subclustering (*Figure 5B*). We integrated the datasets from both the computationally isolated and experimentally purified Mn populations and performed clustering analysis using Seurat. A total of 9144 cells were grouped into 10 clusters (*Figure 5C*). A small cluster (3.6% of total) expressing genes *gfra1a* and *tbx3b* (*Figure 5—figure supplement 1*) was almost entirely sourced from the FACS-sorted datasets (>94%). In addition, it shared markers with a population of non-skeletal muscle Mns recently described in mouse spinal cord scRNAseq datasets (*Blum et al., 2021*). Therefore, this cluster was not included in further comparisons among Mns that control skeletal muscle contraction.

The transcriptionally distinct clusters were next linked to previously known Mn types. Two broad types of Mns have been described for larval zebrafish, the PMn and the SMn, that are commonly distinguished by birth date, progenitor lineage, and morphological features such as size, location, and periphery innervation pattern (*Eisen et al., 1986*; *Myers et al., 1986*; *Menelaou and McLean, 2012*; *Bello-Rojas et al., 2019*). There are four large-sized PMns (CaP, MiP, vRoP, and dRoP) in each hemi-segment of the spinal cord, each innervating approximately one-quarter of axial muscle target field (*Bello-Rojas et al., 2019*; *Wen et al., 2020*). By contrast, 50–70 SMns are in a more ventral location and display a gradient of sizes and functional properties (*Myers et al., 1986*; *Westerfield et al., 1986*; *McLean et al., 2007*; *Asakawa et al., 2012*; *Wang and Brehm, 2017*; *Wen et al., 2020*). We examined the top DEGs among the clusters (*Supplementary file 2*) and found two markers, *chga* and *nr2f1a*, with non-overlapping expression pattern that could reflect this broad classification (*Figure 5C*). *chga* was enriched in one distinct cluster, while *nr2f1a* was present in the majority of the remaining cells (*Figure 5C*). Taken together, these two mutually exclusive markers represented ~95% of Mn population.

In situ hybridization labeling with probes against *chga* and *nr2f1a* revealed spatially segregated Mn groups in the Tg(mnx1:GFP) fish. Specifically, *chga* probes labeled dorsal Mns that were large in size, consistent with them being the primary group (*Figure 5D*). *nr2f1a* labeling was absent in these cells, but was distributed in larger number of smaller Mns that were more ventrally located in the motor column (*Figure 5D*). These segregated patterns of expression strongly suggested that *chga+* cluster represented the PMns, while *nr2f1a* marked the major SMn populations.

For further validation, we used the *chga* probes for in situ hybridization analysis in Tg(SAIG213A; EGFP) fish, in which a single PMn in each hemi-segment, the dorsal projecting CaP, was fluorescently labeled among all the Mns (*Muto and Kawakami, 2011*; *Wen et al., 2020*). Strong *chga* labeling colocalized with the GFP-labeled CaP in each hemi-segment (*Figure 5E*). We also labeled the MiP and RoP Mns in the spinal cord using the sparse labeling approach and observed high level of *chga* signal in both PMn types using in situ hybridization (*Figure 5F*). These results firmly established *chga* as the marker for PMns, leaving the *chga*-negative clusters representing SMns (*Figure 5G*). The number of cells associated with SMn clusters was ~18-fold in excess that of the PMn cluster, consistent with previous cell counts of SMns versus PMns in larval zebrafish (*Eisen et al., 1986*; *Myers et al., 1986*; *McLean et al., 2007*; *Menelaou and McLean, 2012*; *Bello-Rojas et al., 2019*).

## Diversity among SMns

In contrast to the single cluster associated with the PMn type, SMns were composed of multiple tran-scriptionally distinct groups. Approximately 95% of SMns fall into three groups that were distinguished by differential expression of three marker genes, *foxb1b, alcamb,* and *bmp16* (*Figure 6A*). In situ hybridization using the three probes labeled SMns revealed two closely stratified layers that showed differences in dorsal-ventral positioning within the motor column (*Figure 6A*). The *foxb1b+* SMns occupied the more dorsal position of our labeled secondaries while both *alcamb+* and *bmp16+* SMns

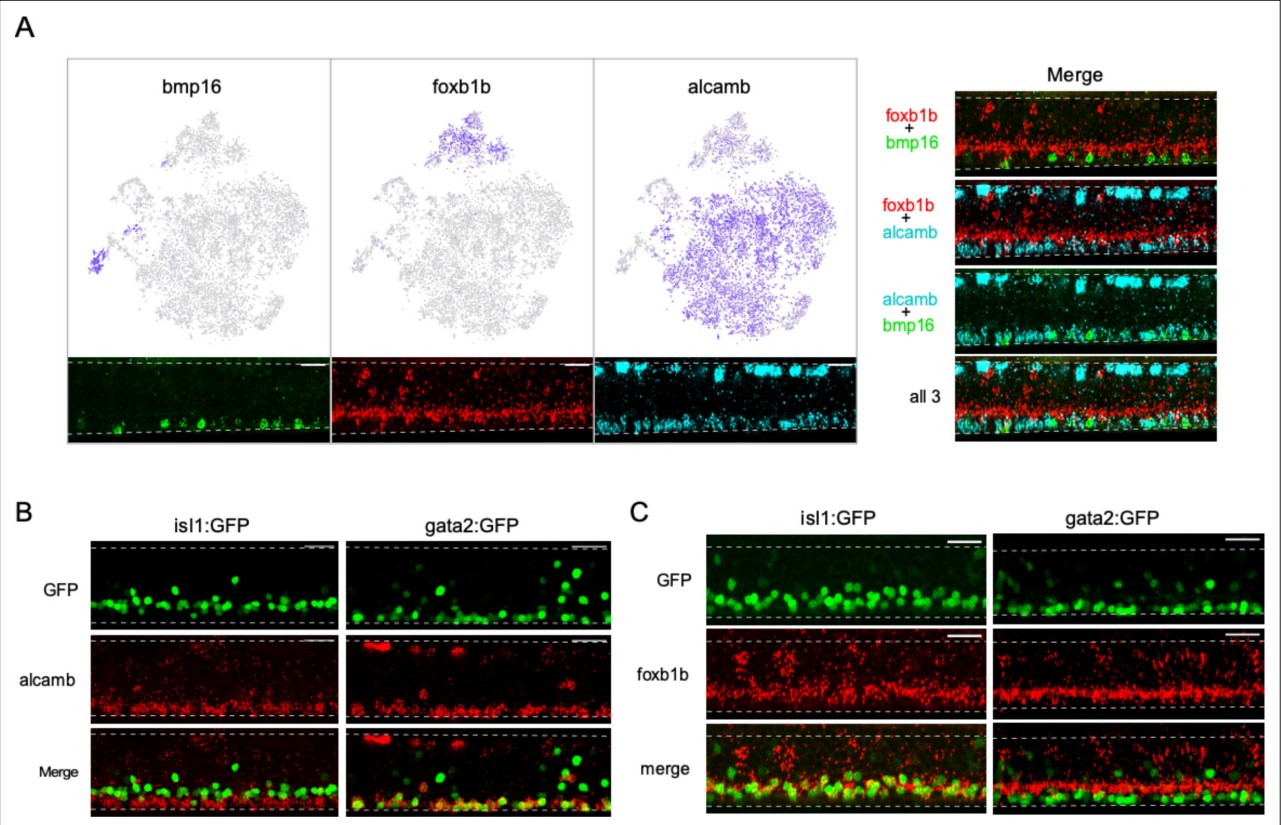

**Figure 6.** Diversity among secondary motoneurons (SMns). (**A**) Differential expression of *bmp16*, *foxb1b,* and *alcamb* associated with distinct SMn clusters. Feature plots in the Mn t-distributed stochastic neighbor embedding (t-SNE) projection (top) and representative in situ hybridization images (bottom) are shown for each marker gene. Merged images of different probe combinations (right) highlight the differences in expression pattern in the motor column. n = 10 fish. (**B**) Representative in situ hybridization images comparing *alcamb* expression in GFP-labeled SMn subpopulations in Tg(isl1:GFP) (left, n = 6 fish) and Tg(gata2:GFP) (right, n = 6 fish). Note that *alcamb* also expresses at high level in the RB neurons located along the dorsal edge of the spinal cord. (**C**) Representative in situ hybridization images comparing *foxb1b* expression in GFP-labeled SMn subpopulations in Tg(isl:GFP) (left, n = 14 fish) and Tg(gata2:GFP) (right, n = 12 fish). Scale bar 20 μm; White dashed line indicates the boundary of spinal cords; dorsal is up.

shared a more ventral location (*Figure 6A*). We next tested *alcamb+* and *foxb1b+* SMns for potential correspondence to the different subsets of SMns labeled with two transgenic lines, Tg(isl1: GFP) and Tg(gata2: GFP) (*Appel et al., 1995*; *Meng et al., 1997*; *Higashijima et al., 2000*). Labeling with in situ probes revealed that expression of *alcamb* overlapped with GFP+ SMns in the Tg(gata2: GFP) fish (*Figure 6B*), while *foxb1b* colocalized with those in Tg(isl1: GFP) (*Figure 6C*). Thus, *foxb1b*/*alcamb* expression distinguishes previously identified SMn subsets. The third cluster, corresponding to the *bmp*-expressing class of SMns, were fewer in number and the targets are not known. However, *bmp* signaling has been linked to specification of slow muscle, raising the possibility that this muscle type receives innervation by *bmp+* Mns (*Kuroda et al., 2013*).

## Transcriptome comparisons for the PMn and SMn types

We performed differential expression analysis comparing PMn and SMn transcriptomes in order to identify candidate genes that might account for their functional distinctions previously established using electrophysiology. After applying thresholds to the Mn clusters, based on average levels of gene expression, p-value, and the proportion of cells expressing individual genes, we obtained a list of 508 candidates that showed at least a 30% difference in average expression levels between Mn types, 317 of which were higher in primaries. To guide our identification of genes that play potential roles in governing excitability and synaptic transmission differences between the two groups of Mns (*Supplementary file 2*), we used Gene Ontology (GO) enrichment analysis for the PMn-specific DEGs (The Gene Ontology Consortium, 2021; *Ashburner et al., 2000*). We focused

on significantly enriched GO terms for DEGs that were associated with three broad categories of biological processes – synaptic function, ion channels/transporters and ion homeostasis, and ATP generation (*Figure 7—figure supplement 1*) – due to their roles in excitability and synaptic transmission. Of the 37 GO terms identified for the PMn DEGs, 24 fell into one of these three broad functionally relevant categories (*Figure 7—figure supplement 1*). We next manually annotated the molecular function of DEGs by referencing evidence-based database ZFIN (*Bradford et al., 2022*) in order to identify specific functionally relevant genes differentially expressed between PMn and SMn. For the PMn type, DEGs encoding a large number synaptic proteins included those of the core exocytotic machinery such as isoforms of VAMP, SNAP25, and Syntaxin; regulators of exocytosis, including Synaptotagmins, NSF, Complexins, and RIM; synaptic vesicle proteins, and synaptic structural proteins such as Synuclein, Bassoon, and Piccolo (*Figure 7A*). Comparisons of expression levels indicate that no synaptic genes were specifically enriched in SMns, but 29 were enriched in PMns compared to SMns (*Figure 7A*). Additionally, five synaptic genes were shared at approximately equivalent levels between Mn types (*Figure 7A*). Similarly, eight different voltage-dependent ion channel subunits were preferentially expressed in the PMn, with the highest levels corresponding to the P/Q calcium channel, the β4 sodium channel subunit, and the Kv3.3 potassium channel. Four other ion channel types were shared by both SMns and PMns. As with the synaptic genes, no ion channel candidates were specifically enriched in the SMn type. The absence of unique ion channel and synaptic gene enrichment in SMns contrasted with the high levels of transcriptional factors and RNA binding proteins, neither of which are likely candidates for conferring functional distinctions between Mn types. Approximately two thirds of the 191 genes enriched in the SMns (118/191) were of these categories. The low levels of candidate ion channels and synaptic genes in the SMns might be expected on the basis of greatly reduced levels of synaptic transmission reflected in quantal content and release probability compared to PMn (*Wang and Brehm, 2017*). These results were consistent even when sorted and unsorted data sources were examined independently (*Figure 7—figure supplement 2*).

Of particular interest among the genes specific to the PMn type were the trio of highest expressing voltage-dependent ion channel types that have been individually linked in previous studies to augmenting transmitter release and/or AP firing rate. Those channel types include a voltage-dependent P/Q-type calcium channel α subunit (*cacna1ab*), a Kv3.3 potassium channel α subunit (*kcnc3a*), and a sodium channel β4 subunit (*scn4ba*) (*Figure 7B and C*; *Eggermann et al., 2011*; *Lewis and Raman, 2014*; *Zhang and Kaczmarek, 2016*). The identification of the *cacna1ab* channel as a top DEG in PMns corroborated previous studies firmly establishing the P/Q-type as the presynaptic active zone calcium channel mediating Ca$^{2+}$-dependent release specifically in the PMns (*Wen et al., 2013*; *Wen et al., 2020*). Enriched expression of the sodium channel β subunit *scn4ba* in PMns was validated by in situ hybridization (*Figure 7D*). *scn4ba* probes specifically labeled dorsally located Mns with large sizes in the Tg(mnx1:GFP) fish (*Figure 7D*). These represented the PMns, as shown by colabeling of individually labeled CaP, MiP, and RoP Mns (*Figure 7D and E*). The zebrafish *kcnc3a* has been shown to be expressed preferentially in PMn at embryonic ages (*Issa et al., 2011*). We tested its expression at 4 dpf fish using immunohistochemical staining. A *kcnc3* subtype-specific antibody efficiently labeled NMJ synaptic terminal marked by α-bungarotoxin (α-Btx, *Figure 7F2*). Since the PMn and SMn axons track along with each other and form synapses with the same postsynaptic receptor clusters (*Wen et al., 2020*), further resolution was needed to distinguish the PMn and SMn synaptic terminals. For this purpose, we ablated individual CaPs by transiently expressing the phototoxic KillerRed protein (*Formella et al., 2018*), followed by light inactivation at 2 dpf. By 4 dpf, the ablation of CaP was complete as indicated by the absence of its soma and processes (*Figure 7F1*). *Kcnc3* antibody staining showed a specific signal reduction in synapses located in the ventral-most musculature (*Figure 7F2*), the target field shared by the CaP and numerous other SMns. This result strongly supports the expression specificity of *kcnc3a* channel in the PMns.

These specific calcium, potassium, and sodium channel isoforms coexpress specifically among the four types of PMns. We term this collective unit of three different voltage-dependent ion channel types as a 'channel cassette.' Over 55% of the cells in PMn cluster express the cassette compared to <3% in the SMns over all (*Figure 7C*). Together with other synaptic DEGs revealed by the analysis, our results suggested a molecular logic for the functional specialization in PMn synapses, such as strong release and high-frequency firing, that are uniquely associated with escape behavior.

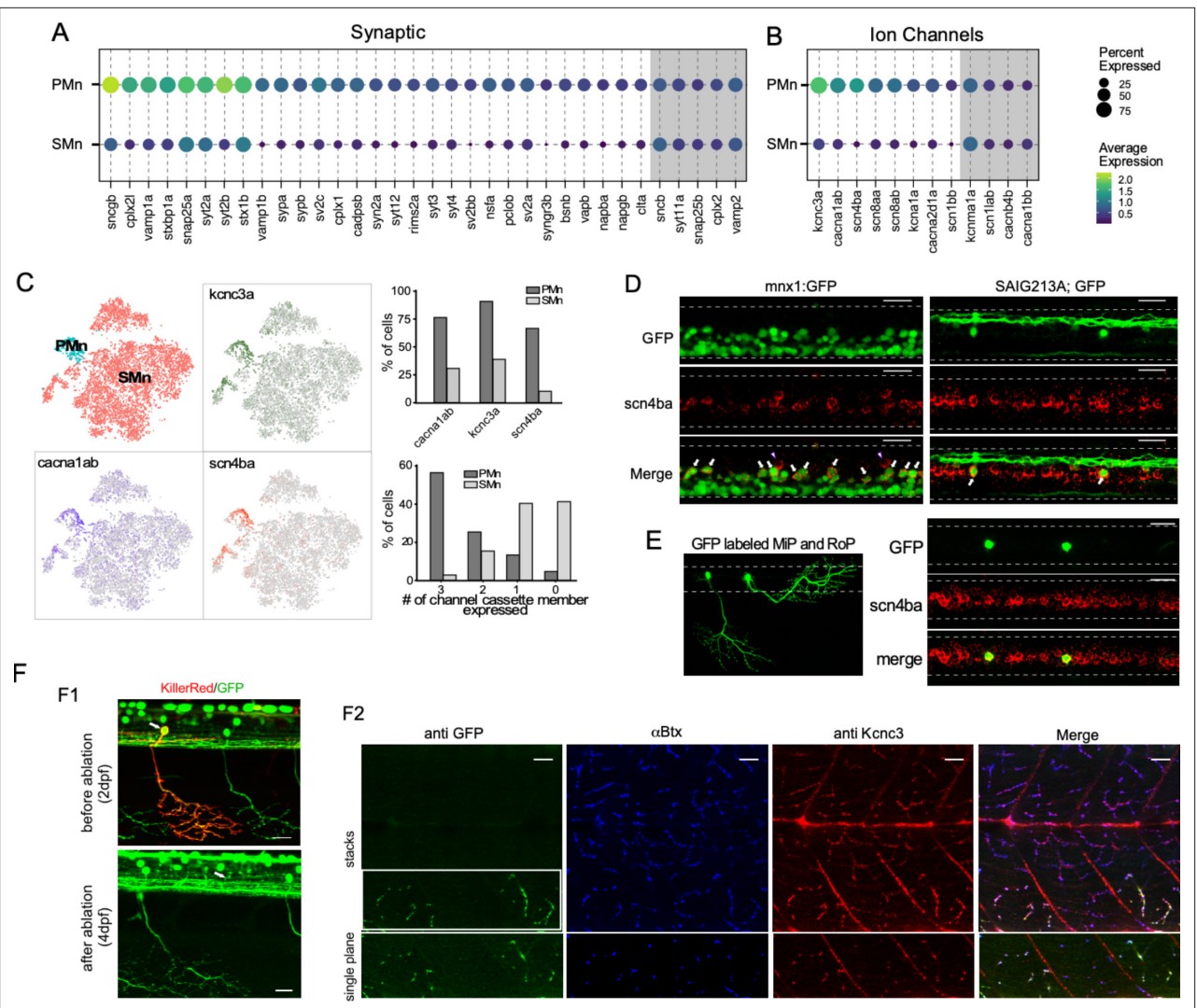

**Figure 7.** Transcriptome comparison between primary motoneurons (PMns) and secondary motoneurons (SMns). (**A**) Dot plot for synaptic genes differentially enriched in PMns compared to SMns. Both percentage of cells with expression and average expression levels are shown. Examples of synaptic genes that are expressed at comparable levels between the two Mn types are shaded in gray. (**B**) A similar comparison for differentially expressed ion channel genes as shown in (**A**). (**C**) Feature plots for three top differentially expressed ion channel genes, *cacna1ab*, *scn4ba*, and *kcna3a*, shown in the Mn t-distributed stochastic neighbor embedding (t-SNE) projection (left). The assignment of Mn type identity was duplicated from *Figure 5G* for reference (right graphs). The proportion of cells in each Mn type expressing individual cassette member (top) and cassette member combinations (bottom). (**D**) Representative in situ hybridization images with *scn4ba* probes in Tg(mnx1:GFP) (left) and Tg(SAIG213A;GFP) (right) transgenic fish. Each image shows approximately two segments of the spinal cord in the middle trunk of 4 days post fertilization (dpf) fish. Arrows indicate the PMns in Tg(mnx1:GFP) and CaP in Tg(SAIG213A;GFP) fish (n = 15–18 fish). Two commissural local (CoLo) interneurons labeled with *scn4ba* probes are also indicated (arrowhead). (**E**) Expression of *scn4ba* in the MiP and RoP PMns. The morphology of GFP-labeled MiP and RoP in an injected fish shown (left). In situ hybridization images with *scnba* probes in this fish showed colocalization with GFP labeling (right). n = 7–10 cells. Scale bar 20 µm; white dashed line indicates the boundary of spinal cord; dorsal is up. (**F**) Validation of *kcnc3a* enrichment in PMns by immunohistochemistry staining. (**F1**) KillerRed-mediated photoinactivation of CaP. Representative fluorescent images showing approximately two segments of a Tg(SAIG213A;EGFP) fish with a single CaP (arrow) expressing KillerRed, before photo illumination at 2 dpf (top), and ~40 hr after inactivation (bottom). Both the soma (the location indicated by an arrow) and periphery branches (see also **F2** leftmost panel) are absent after the ablation. (**F2**) Immunohistochemical staining of the same fish with a Kcnc3-specific antibody. GFP expression is revealed by anti-GFP antibody staining, and the location of synapses labeled by α-Btx. Top panels represent a maximal intensity projection of a stacked of z-plane images, while the bottom shows a single focal plane of the CaP target field (indicated by a white box). Scale bar 20 µm. n = 5 fish.

The online version of this article includes the following figure supplement(s) for figure 7:

**Figure supplement 1.** Gene Ontology (GO) analysis for primary motoneuron (PMn) differentially expressed genes (DEGs).

**Figure supplement 2.** Combining motoneuron (Mn) datasets from two isolation methods.

## Gene ensembles for additional components of escape circuitry

Further insights into the potential contribution the ion channel cassette to fast synaptic function specialization came from examination of their differential expression pattern in other circuitry components involved in high-speed swimming and escape. Those cell types include the CoLo inhibitory and v2a excitatory interneurons, both of which provide direct synaptic connections to the PMn type. Both type I and type II v2a neurons participate in high swim speeds, but the type II is more effectively recruited over the range of higher speeds (*Hale et al., 2001*; *Higashijima et al., 2004c*; *Kimura et al., 2006*; *Liao and Fetcho, 2008*; *McLean et al., 2008*; *Satou et al., 2009*; *Ampatzis et al., 2014*; *Menelaou and McLean, 2019*; *Satou et al., 2020*). Additionally, electrophysiological studies have shown that type II v2a neurons fire faster and more reliably than type I v2a neurons (*Menelaou and McLean, 2019*). As seen in the PMn type, the coexpression of the ion channel cassette is high in both type II v2a and CoLo interneuron types compared to other interneuron types (*Figure 8A*). Some of those interneuron types expressed individual components of the cassette, most often *kcnc3a* or *cacna1ab* channel types, but not the full cassette. Indeed, the *scn4ba* sodium channel β subunit is the strongest delineator among the three for inclusion into the ion cassette classification (*Figures 7C and 8A*).

Top DEGs identified for the PMns that encode synaptic proteins also showed higher levels of expression in type II v2a and CoLo interneurons compared to other cell types, including *vamp1a, vamp1b, syt2a, syt2b, snap25a, stxbp1a,* and *cplx2l*, all central players in transmitter release (*Figure 8B*). This further strengthens the proposition that a synaptic cassette works collectively with the ion channel cassette to create a strong neuronal circuitry controlling the escape behavior and fast swimming behavior. These results are incorporated into a simplified model that potentially accounts for the differential circuitry that distinguish regulation of slow swimming from the power circuits (*Figure 8C*).

## Discussion

Interest in the spinal circuitry controlling muscle movements among vertebrates remains high, especially from the standpoint of understanding inheritable human myasthenic disorders (*Walogorsky et al., 2012b*; *Wen et al., 2016a*; *Ono et al., 2002*; *Wang et al., 2008*; *Downes and Granato, 2004*). As a model for investigation into spinal circuits, zebrafish offers great simplicity when linking spinal circuitry to a specific behavior. Moreover, morphological, physiological, and transgenic labeling techniques have revealed a large repertoire of spinal neuronal types involved in locomotion in zebrafish, many of them sharing homologies with those in mouse (*Grillner, 2003*; *Goulding, 2009*; *Grillner and Jessell, 2009*). In both mouse and larval zebrafish, the understanding of motility circuitry centers on the study of Mns, which are the final pathway to movement. Unlike the numerous Mn subtypes in mammals, which are grouped according to different anatomical positions and muscle targets (*Stifani, 2014*), there are two main classes in zebrafish that share the same fast muscle cells as well as the individual neuromuscular synapses (*Eisen et al., 1986*; *Myers et al., 1986*; *Menelaou and McLean, 2012*; *Bello-Rojas et al., 2019*; *Wen et al., 2020*). To aid in the assignment of neuronal types to specific circuits, we performed scRNAseq analysis on larval zebrafish spinal cord. Our study revealed new markers for key components of the spinal circuitry that likely contribute to regulation of different behavioral responses, along with identification of a new interneuron type.

Studies on Mn control of movement in zebrafish have focused on two idiosyncratic swimming behaviors; the single powerful tail bend initiating escape and the subsequent, generally less powerful, rhythmic swimming (*Budick and O'Malley, 2000*; *Thorsen et al., 2004*). The spinal circuits mediating these two behaviors have been the subject of many studies, which conclude that the SMns do not play important roles in behavioral escape but instead regulate swimming over a broad range of speeds. Consistent with this role, they cover a lower range of AP firing and release much less transmitter per AP compared to the PMn type (*McLean et al., 2007*; *Wang and Brehm, 2017*; *Wen et al., 2020*). Thus, PMns regulate the highest speed swimming and escape behavior. Indeed, the functional distinctions were our impetus to identify individual transcriptomes for the separate Mn types. Using a newly identified marker, *chaga*, we identified a single small cell cluster corresponding to the four PMns located within each hemi-segment. The SMns, by contrast, corresponded to three different clusters based on the markers *foxb1b, alcamb,* and *bmp16*. These three transcripts were used to localize SMn subtype populations within the spinal cord. The SMn somas containing *foxb1b* transcripts formed the

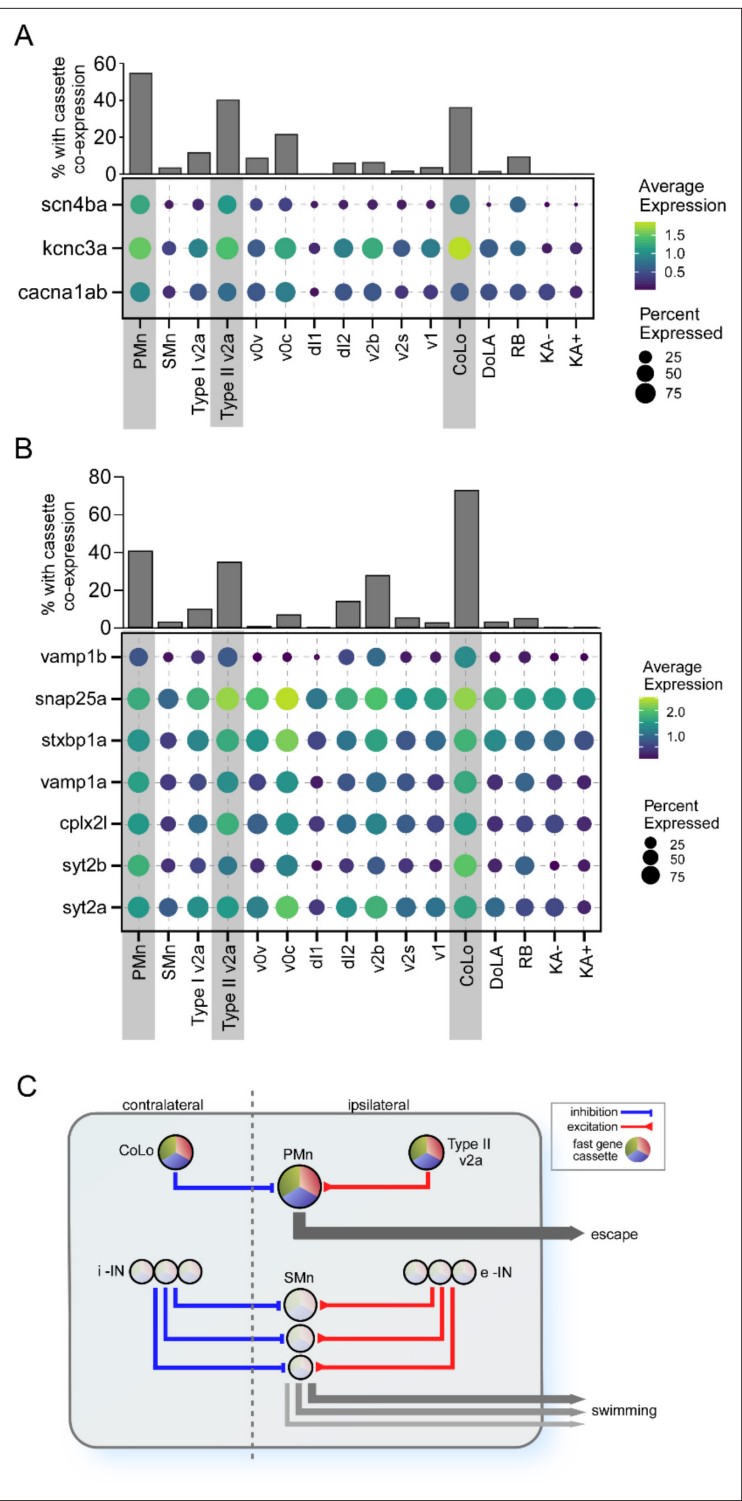

**Figure 8.** Differential expression of gene cassettes in larval zebrafish escape circuit. (**A**) The ion channel cassette. (Lower) Dot plot showing the averaged expression level (color scale) and percentage of cell expressed (dot area) of *scn4ba*, *kcnc3a*, and *cacna1ab* channel genes in different neuronal types. The three neuronal types located at the escape circuit output pathway are highlighted (gray shade). (Upper) Bar graph showing the percentages of cells in each neuronal type coexpressing all three channel genes. (**B**) The cassette of synaptic genes. Seven top primary motoneuron (PMn) differentially expressed genes (DEGs) encoding proteins involved in synaptic transmission are shown for all neuronal types. (**C**) Proposed circuitry for separate control over escape and swimming in larval zebrafish. The schematic model is based on published studies and incorporates the role of the DEG cassettes in

*Figure 8 continued on next page*

*Figure 8 continued*

conferring behavioral and functional distinctions that are manifest both centrally and at the NMJ. The circuitry and cassette expression in the PMn that control escape are illustrated at the top, and the SMn circuitry that controls swim speed is illustrated at the bottom. Swim speed is dependent on Mn size as published, which is determined at the levels of both spinal circuitry and neuromuscular synaptic strength. According to this simplified model, the gradient of synaptic strength and speed at the NMJ is set by the levels of cassette expression among Mns. i-IN, inhibitory interneurons; e-IN, excitatory interneurons.

most dorsal group, which are positioned above the *alcamb*-labeled group in the motor column. The smallest cluster, labeled by *bmp16*, overlapped with the *alcamb* label. The relative position of these SMn clusters is consistent with studies showing a correspondence between the dorsal-ventral position within the spinal cord and control of rhythmic swim speed by the SMns. Sequential recruitment of more dorsally located SMns leads to the generation of increased power and faster swim speed (*McLean et al., 2007*; *Gabriel et al., 2011*; *Wang and Brehm, 2017*). This gradient of power is determined through both firing pattern and by amount of transmitter release at the NMJ (*Wang and Brehm, 2017*). Thus, it is plausible that the transcriptomic distinctions among SMns reflect their differential roles in swim speed determination and their functional distinctions in synaptic strength (*Wang and Brehm, 2017*; *Wen et al., 2020*).

Previous electrophysiological studies indicate that PMns fire APs at a high frequency and that transmitter release occurs with a short synaptic delay, high quantal content, and a high release probability (*Wen et al., 2016b*). By contrast, SMn synapses are much weaker and variable in firing properties, in keeping with the distinct behavioral roles of PMns and SMns (*Wang and Brehm, 2017*; *Wen et al., 2020*). Consistent with these distinctions, analysis of the transcriptomes for the two Mn types revealed large-scale differences. The PMns expressed high levels of transcripts encoding ion channels and exocytotic machinery, all of which are generally not detected in SMns. Overall, the transcriptomic profile comparisons are consistent with the electrophysiological findings of greatly enhanced neuromuscular transmission for the PMns. In particular, the PMns had very high expression of an ion channel cassette formed by three different voltage-dependent ion channel types, each of which had been linked previously to either high release probability of transmitter or very high AP frequency. The cassette transcript with the most restrictive expression pattern encoded the β4 subunit of the voltage-dependent sodium channel NaV1.6. This subunit confers high-frequency firing through a fast reversible block of the NaV1.6 channel pore. The blocking kinetics are sufficiently fast to enable the neuron to fire APs at frequencies exceeding the refractory period (*Raman and Bean, 1997*; *Grieco et al., 2005*; *Lewis and Raman, 2014*; *Ransdell et al., 2017*). A second highly enriched voltage-dependent channel in PMns is the Kv3.3 potassium channel. This ion channel type has been associated with high AP firing frequency, as well as with augmented transmitter release in mammalian neurons (*Zhang and Kaczmarek, 2016*; *Richardson et al., 2022*), both due to fast activation kinetics that result in fast repolarization of the AP. The zebrafish *kcnc3a* gene, encoding the Kv3.3 channel, gives rise to a transient potassium current that has both fast activation and inactivation, making it well suited for its proposed role in shortening AP waveform and allowing high-frequency firing (*Mock et al., 2010*). The third cassette member, *cacna1ab*, encoding a P/Q- type calcium channel, was a top DEG in the PMn, in agreement with our previously published finding that the SMn expresses a different calcium channel isoform, most likely N-type based on sensitivity to specific conotoxin isoforms (*Wen et al., 2020*). Mutations in the *cacna1ab* gene completely abolished AP-evoked release in PMns but left synaptic transmission in SMns intact (*Wen et al., 2020*). As a result, mutant fish are unable to mount a fast escape response, but are still capable of normal fictive swimming (*Wen et al., 2020*). The P/Q-type calcium channel has been associated widely with synapses with high release probability (*Iwasaki and Takahashi, 1998*; *Wu et al., 1999*; *Stephens et al., 2001*; *Fedchyshyn and Wang, 2005*; *Bucurenciu et al., 2010*; *Eggermann et al., 2011*; *Young and Veeraraghavan, 2021*), a feature due in part to a higher open probability during APs compared to the N-type counterparts (*Li et al., 2007*; *Naranjo et al., 2015*), thereby promoting calcium entry. We hypothesize that the collective actions of this ion channel cassette, with their unique biophysical properties, serve to mediate the escape response and the highest swim speed through ultrafast repetitive firing and maximal release of neurotransmitter. This trio of voltage-dependent ion channels is also coexpressed in mammalian neuronal types that are involved in fast signaling, suggesting a conserved role for the cassette in fast behaviors. Examples

include the auditory neuron calyx of Held (*Iwasaki and Takahashi, 1998*; *Midorikawa et al., 2014*; *Richardson et al., 2022*; *Kim et al., 2010*) and the pyramidal neurons of the brain (*Grieco et al., 2005*; *Akemann and Knöpfel, 2006*; *Hillman et al., 1991*).

A second cassette comprised of a set of genes involved in synaptic function was also revealed by comparing the transcriptomes between the PMns and SMns. The cassette components enriched in PMns were genes encoding isoforms of VAMP, Syntaxin, Synaptotagmin, SNAP25, and Complexin, all components associated with exocytosis and transmitter release. Unlike the ion channel cassette described above, which is strongly linked to synapses with high release probability and/or high firing rate, the actions by which these synaptic genes could differentially support strong and fast synaptic properties remain speculative. It has been suggested, however, at both fly NMJ and mammalian CNS, which the differential synaptic strength among synapses correlates with abundance of proteins involved in transmitter release (*Holderith et al., 2012*; *Peled et al., 2014*; *Akbergenova et al., 2018*).

Further support for the idea that ion channel and synaptic cassettes both play direct roles in the formation of specialized circuitry surrounding the PMn was provided by transcriptomic analysis of interneuron types known to interact specifically with the PMn and to be recruited during high-speed swimming. Those include the type II v2a excitatory interneuron and CoLo inhibitory interneuron forming the output pathway for the escape response (*Bhatt et al., 2007*; *Liao and Fetcho, 2008*; *Satou et al., 2009*; *Menelaou and McLean, 2012*; *Menelaou and McLean, 2019*). As with the PMn, both interneuron types coexpress the triple ion channel cassette members at high levels compared to those interneurons that participate less during recruitment during high-speed swimming. As shown for distinction among Mn types, the sodium channel β4 appears to be the most restrictive among the three ion channel types in conferring fast firing to the interneurons as well. Unlike the case for the PMn, the AP firing and transmitter release properties of these neuronal types are less well established. However, it is clear that the type II v2a, in particular, can fire at high frequencies over 600 Hz (*Menelaou and McLean, 2019*). The highly specific enrichment of these two gene cassettes, in neurons involved in escape behavior, suggests that they serve as an integral part of the molecular signature underlying functional specialization. It remains to be seen whether the gene cassettes we identified for zebrafish escape circuit represent a general transcriptional architecture plan to build synapses with great strength and speed in the CNS of higher vertebrates.

Finally, our scRNAseq analyses provides a resource for future identification of gene functions that are causal or associated with human disorders involving Mn dysfunction. In the context of myasthenic disorders in particular, zebrafish has provided a large number of animal models corresponding to human syndromes, including slow channel syndrome, episodic apnea, and rapsyn deficiency (*Ono et al., 2002*; *Wang et al., 2008*; *Walogorsky et al., 2012a*; *Walogorsky et al., 2012b*; *Wen et al., 2016a*). Both myasthenic syndromes and amyotrophic lateral sclerosis (ALS) involve dysfunction at the level of the motor circuits (*Ferraiuolo et al., 2011*). In the case of ALS, it is well known that fast motor neurons are selectively targeted for degeneration (*Hadzipasic et al., 2014*; *Nijssen et al., 2017*). Our analysis comparing transcriptional profiles between fast versus slow motor circuit components offers a new means for probing the transcriptional consequences of neuromuscular disease states.

## Materials and methods

### Fish lines and husbandry

The transgenic line Tg(mnx1:GFP) was provided by Dr. David McLean (Northwestern University). Tg(vsx2:Kaede) was provided by Dr. Joseph Fetcho (Cornell University). Tg(SAIG213A;EGFP), Tg(islet1:GFP) and Tg(gata2:GFP) were maintained in the in-house facility. Zebrafish husbandry and procedures were carried out according to the standards approved by the Institutional Animal Care and Use Committee at Oregon Health & Science University (OHSU) (IP00000344). Experiments were performed using larva at 4 dpf. Sex of the larva cannot be determined at this age.

### Sparse labeling of spinal neurons

In most cases, we used the Gal4-UAS system to achieve mosaic expression by co-injection of two plasmids into single-cell embryos: one containing Gal4 driven by cell type-specific promoters and the other containing fluorescent reporter genes under the control of UAS element. The mnx1 or vAChT promoter drove the expression in Mns (mnx1:Gal4 plasmid provided by Dr. McLean, Northwestern

University, and vAChT:Gal4 provided by Dr. Joe Fetcho, Cornell University). drmt3a promoter (drmt3a:Gal4 provided by Dr. Shinichi Higashijima, National Institute of Natural Sciences, Japan) was used for expression in glycinergic inhibitory interneurons. mCherry driven by HuC promoter was used to label spinal neurons, including the DoLA interneurons. Injected fish were screened on 3 dpf for sparse fluorescent neurons that could provide detailed morphology.

## KillerRed-mediated CaP ablation

We transiently expressed the phototoxic KillerRed protein in Tg(SAIG213A:EGFP) fish by injecting a plasmid expressing KillerRed driven by UAS promoter (Addgene plasmid #115516; a gift from Marco Morsch). Fish with KillerRed expression in CaP were identified. Individual CaPs were ablated by light inactivation at 2 dpf by illuminating for 10 min with a 560 nm laser set at high power. This completely bleached KillerRed fluorescence and induced visible blebbing in CaP terminals. Fish were grown to 4 dpf, and the ablation was confirmed by the absence of GFP-labeled soma and neurites.

## Whole-mount immunocytochemistry

Whole-mount immunohistochemistry was performed as described previously (*Wen et al., 2020*). Also, 4 dpf larvae were fixed in 4% paraformaldehyde at 4°C for 4 hr. Zebrafish Kcnc3a channel was labeled using a polyclonal antibody originally generated against the human Kcnc3 protein (Thermo Fisher Scientific PA5-53714) at a concentration of 2.5 µg/ml. Polyclonal anti-GFP (Abcam ab13970) was used at 1 µg/ml. Alexa Fluor-conjugated secondary antibodies (Thermo Fisher Scientific) were used at 1 µg/ml. To mark the location of synapses, 1 µg/ml CF405s-conjugated α-Btx (Biotium) was included in the secondary antibody incubation to label postsynaptic acetylcholine receptors.

## Whole-mount in situ RNA hybridization

Fluorescence in situ RNA hybridization (FISH) was performed on whole-mount 4 dpf larva using the multiplexed hybridization chain reaction RNA–FISH bundle (HCR RNA-FISH) according to the manufacturer's instructions (Molecular Instruments; *Choi et al., 2016*). Probe sets for zebrafish *alcamb*, *nr2f1a*, *chga*, *scn4ba*, *foxb1b*, *bmp16*, *gjd2b*, and *pnoca* were custom-designed based on sequences (Molecular Instruments) and used at 4 nM each. Fluorescent HCR hairpin amplifiers were used at 60 nM each to detect the probes. GFP and mCherry fluorescence in the transgenic lines and transient labeled neurons survived the FISH protocol with signal loss mostly limited to the periphery. Residual fluorescence was sufficient to mark the location of soma in the spinal cord without the need for additional amplification.

## Fluorescence imaging

After staining, fixed larval were mounted in 1.5% low melting agarose and imaged on a Zeiss 710 laser-scanning microscope equipped with an LD C-Apochromat ×40/1.2 n.a. objective. Z-stacks of confocal images were acquired using Zen (Carl Zeiss) imaging software and presented as either maximal intensity projection or single focal planes as indicated in the figures (ImageJ, National Institutes of Health).

## Single-cell suspension from spinal cord for scRNAseq

Single-cell suspensions were prepared from Tg(SAIG213A;EGFP) fish for the two full spinal cord datasets and from Tg(mnx1:GFP) for the two FACS-sorted Mn enrichment datasets.

About 150 4 dpf larva were euthanized in 0.02% tricaine and individually decapitated behind the hindbrain. They were incubated with 20 mg/ml collagenase (Life Sciences) in a buffer containing 134 mM NaCl, 2.9 mM KCl, 1.2 mM MgCl₂, 2.1 mM CaCl₂, and 10 mM Na-HEPES (pH 7.8) at 28°C for 2 hr, with intermittent trituration using a p200 pipette aid at 0, 0.5 hr, and 1 hr of the incubation. To release spinal cords from remaining tissue, the final triturations were done using fire-polished Pasteur pipettes with decreased opening sizes (300, 200, and 100 µm, respectively). Intact spinal cords were transferred to L15 media and washed three times with fresh media. The spinal cords were incubated with 0.25% trypsin solution (in 1× PBS containing 1 mM EDTA) at 28°C for 25 min. The digestion was terminated by adding 500 µl stop solution (L15 with 1% fetal bovine serum). The tissue was collected by spinning at $400 \times g$ for 3 min at 4°C, washed once with L15, and resuspended in 200 µl of L15 media. Spinal cord cells were dissociated by triturating the digested tissue with fire-polished Pasteur pipettes with 80–100 µm opening. The solution was filtered through a 35 µm strainer into a siliconized

collection tube. The suspension was examined on a microscope for cell count, Trypan blue staining-based viability test, and proportion of dispersed single cells. Samples with a viability above 70% were used for sequencing.

## FACS sorting

Single-cell suspension prepared from Tg(mnx1:GFP) fish were FAC-sorted for EGFP+ cells using a 100 µm nozzle on a BD inFlux cell sorter (Flow Cytometry Shared Resource, OHSU). Cells were collected in 100 µl PBS containing 0.2% bovine serum albumin in a siliconized tube.

## Single-cell capture, cDNA synthesis, and library preparation and sequencing

Single-cell capture, cDNA synthesis, and library preparation were performed by the Massive Parallel Sequencing Shared Resource at OHSU using the 10X Genomics Chromium v3.0 reagent kit. Single-cell suspension for the two full spinal cord replicates targeted 10,000–15,000 cells. For the two FACS-sorted samples, 4000–5000 cells were targeted. Replicate samples were prepared from different clutches of animals. Libraries were sequenced on an Illumina NovaSeq 500 instrument to an average read depth of ~40,000 per cell.

## Reference genome generation and alignment

Cellranger v6.1.1 (10X Genomics) was used for the reference genome generation and alignment. Our reference genome was generated by modifying a preexisting reference genome, Lawson v4.3.2 (*Lawson et al., 2020*), which we edited to add an EGFP sequence as an artificial chromosome and to correct a selection of gene names and 3′ untranslated region (UTR) annotations. A full account of all changes made to the Lawson reference genome can be found in *Supplementary file 3*, with a representative example shown (*Figure 1—figure supplement 1*). Alignment to this reference genome was performed using Cellrangers count function with expect-cells set to the targeted number of cells for each replicate.

## Preprocessing, normalization, clustering analysis, and visualization

Count matrices were processed using the Seurat v4.0 package for R (https://www.satijalab.org/seurat/; *Hao et al., 2021*). Genes with expression in less than three cells were excluded from further analysis. Initial quality control was performed on each sample independently. Cells were kept for further analysis if they had a number of unique genes between 400 and 4000, UMI counts between 1500 and 9000, and <5% mitochondrial gene content. No further doublet removal methods were applied. Data was normalized using the Seurat Sctransform v2 package in R, generally following the procedure outlined in the Introduction to SCTransform v2 regularization vignette (*Hafemeister and Satija, 2019*; *Choudhary and Satija, 2022*). Principal components were calculated, followed by nearest-neighbor graph calculation using ANNoy implemented through Seurat. Clustering used the Leiden community detection method implemented with the FindClusters function and the Leidenalg Python package (*Traag et al., 2019*). To visualize clustered datasets, we used t-distributed stochastic neighbor embedding (t-SNE) (*Maaten and Hinton, 2008*) or Uniform Manifold Approximation Projection (UMAP) (*McInnes et al., 2018*) implemented through Seurat.

## Dataset integration

Datasets were combined using Seurat's integration pipeline (*Stuart et al., 2019*), considering 9000 variable genes. To examine the correspondence between duplicates, exploratory clustering was done in the combined datasets (*Figure 1—figure supplement 2A and B*). Intermixing of data points from different samples was inspected both visually and by plotting the distribution against one another (*Figure 1—figure supplement 2A and B*). For the two full spinal duplicates, one cluster of cells exhibited a strong bias towards a single replicate, with over 90% of the cells sourcing from a single replicate (*Figure 1—figure supplement 2A*). These cells were not included in the combined dataset for downstream analysis to control for technical and biological variability. The FACS-sorted duplicates had no obvious outliners (*Figure 1—figure supplement 2B*).

The combined motor neuron dataset was generated by integrating Mns extracted from the full spinal dataset and the FACS-sorted dataset, based on expression of canonical Mn markers (*Figure 5A*

*and B*). While the relative populations of SMns did vary between the two sample sources, there was no Mn subtypes that could not be identified independently in both methods of Mn isolation (*Figure 7—figure supplement 2*). Differential expression analysis performed independently with either Mn source yielded highly reproducible sets of the DEGs (*Figure 7—figure supplement 2*), further validating the results from the combined dataset.

## Annotation of cell clusters

Specific cell types in the neuronal subset of the spinal data were annotated on the basis of significantly differentially expressed genes. Exploratory clustering analysis was first conducted to filter out cells not of spinal cord origin. One small cluster in the whole spine dataset (0.6% of cells) expresses *tph2/ucn3l/slc18a2* at high level compared to all the rest of the clusters (*Figure 1—figure supplement 3*), marker genes for serotonergic raphe nucleus in the hindbrain (*Oikonomou et al., 2019*; *Bradford et al., 2022*). They reflected a trace amount of hind brain tissue during spinal cord dissection, and were removed from the analysis. Contaminating muscle cells were also removed based on expression of *my1pfa/tnnt3b/actc1b* myosin/troponin/actin genes (*Figure 1—figure supplement 3*). Overall, these contaminants accounted for ~2.1% of the total cell population isolated from the spinal cord. No cells in the FACS-sorted dataset were identified as originating from contamination from outside the spinal cord.

## Data analysis and statistical tests

Differential gene expression was calculated using a Wilcoxon rank-sum test implemented with the FindMarkers or FindAllMarkers functions in Seurat. Genes were considered to be enriched in a cluster if they had a log2 fold change >0.38 (corresponding to >30% enrichment in expression level), expression in at least 30% of cells in one of the groups being compared, and a Bonferroni adjusted p-value of <$10^{-5}$. A more conservative adjusted p-value of <$10^{-10}$ was used for comparison between PMns and SMns.

## GO analysis

DEGs comparing PMns and SMns were used to generate lists of GO terms enriched in the PMns using the EnrichR package in the aspect of 'biological processes' (*Chen et al., 2013*; *Kuleshov et al., 2016*; *Xie et al., 2021*). GO terms were considered significantly enriched with adjusted p-value<0.05.

## Acknowledgements

The authors thank Drs. Apiar Saunders and Alex Nechiporuk for advice with the scRNAseq analysis, Massive Parallel Sequencing Shared Resource at OHSU, for single-cell capture, cDNA synthesis and library preparation and sequencing, and Flow Cytometry Shared Resource at OHSU for FACS sorting. Kara Grist provided expert zebrafish husbandry. This research was funded through a grant from the NIH (NS105664) to PB.

## Additional information

### Funding

| Funder | Grant reference number | Author |
| --- | --- | --- |
| National Institutes of Health | NS105664 | Paul Brehm |

The funders had no role in study design, data collection and interpretation, or the decision to submit the work for publication.

### Author contributions

Jimmy J Kelly, Resources, Data curation, Software, Formal analysis, Visualization, Writing - review and editing; Hua Wen, Conceptualization, Formal analysis, Validation, Visualization, Methodology, Writing

- original draft, Writing - review and editing; Paul Brehm, Funding acquisition, Investigation, Writing - original draft, Writing - review and editing

## Author ORCIDs

Jimmy J Kelly ⓘ http://orcid.org/0009-0007-5807-0676
Hua Wen ⓘ http://orcid.org/0009-0009-5326-8741
Paul Brehm ⓘ http://orcid.org/0000-0001-7804-5258

## Ethics

This study was performed in accordance with the recommendations in the Guide for the Care and Use of Laboratory Animals by the National Institutes of Health. All of the animals were handled according to approved institutional animal care and use committee (IACUC) protocols (IP00000344) of the Oregon Health and Science University.

Reviewer #1 (Public Review): https://doi.org/10.7554/eLife.89338.3.sa1
Reviewer #2 (Public Review): https://doi.org/10.7554/eLife.89338.3.sa2
Reviewer #3 (Public Review): https://doi.org/10.7554/eLife.89338.3.sa3
Author Response https://doi.org/10.7554/eLife.89338.3.sa4

---

# Additional files

## Supplementary files

• Supplementary file 1. Table of differentially expressed genes (DEGs) in spinal neuron clusters.

• Supplementary file 2. Table of differentially expressed genes (DEGs) in primary motoneurons (PMns) versus secondary motoneurons (SMns).

• Supplementary file 3. Changes to the Lawson v4.3.2 reference genome.

• MDAR checklist

## Data availability

Raw fastq files and unprocessed aligned data can be accessed for free through the Gene Expression Omnibus (accession number GSE232801). All code used in data processing and figure creation has been deposited, and is freely available on github (https://github.com/JimmyKelly-bio/Single-cell-RNA-seq-analysis-of-spinal-locomotor-circuitry-in-larval-zebrafish, copy archived at *Kelly-bio, 2023*).

The following dataset was generated:

| Author(s) | Year | Dataset title | Dataset URL | Database and Identifier |
|---|---|---|---|---|
| Kelly JJ, Wen H, Brehm P | 2023 | sc-RNAseq of the day 4 zebrafish spinal cord | https://www.ncbi.xyz/geo/query/acc.cgi?acc=GSE232801 | NCBI Gene Expression Omnibus, GSE232801 |

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
