## [Editor Report · eLife assessment]

In zebrafish, primary motor neurons (PMNs) control escape movements, and a more heterogeneous population of secondary motor neurons (SMNs) regulate the speed of rhythmic swimming. Using single-cell RNA sequencing (scRNAseq), the authors have obtained **compelling** evidence that PMNs, and two types of interneurons innervating them, express a set of three genes encoding voltage-gated ion channels enabling rapid firing. The PMNs also express high transcript levels of proteins involved in exocytosis, which would be expected to support rapid neurotransmitter release. These results will be **important** for those working on spinal cord function and zebrafish genomics/transcriptomics.

---

## [Referee Report · Reviewer #1 (Public Review)]

This manuscript by Kelly et al. reports results from single-cell transcriptomic analysis of spinal neurons in zebrafish. The work builds on a strong foundation of literature and the objective, to discern gene expression patterns specializing functionally distinct motor circuits, is well rationalized. Specifically, they compared the transcriptomes in the escape and swimming circuits.

The authors discovered, in the motor neurons of the escape circuit, two functional groups or "cassettes" of genes related to excitability and vesicle release, respectively. Expression of these genes make sense for a "fast" circuit. This finding will be important to the field and form the basis for subsequent studies differentiating the escape circuit from others.

---

## [Referee Report · Reviewer #2 (Public Review)]

Summary: Kelly et al. strategically leverage the unique strengths of the zebrafish larval model and scRNA-seq to uncover genes that determine the stereotypic output of different neuronal circuits. The results lead to the identification of ion channel and synapse associated genes that distinguish a fast reliable neuronal circuit.

Strengths:

- Well-established neuronal markers allow the transcriptomic analyses to match a majority of the transcriptomic clusters to specific spinal neuron subtypes.

- One transcriptomic cluster reveals the presence in zebrafish of a spinal neuron subtype previously identified in mammals.

- The primary motor neuron and specific interneurons of the circuit mediating strong and fast swimming share expression of cassettes of ion channel and synapse-related gene cassette that sculpt fast and strong synaptic transmission.

- Results are optimally placed in the context of the rich background and literature regarding zebrafish spinal neuron physiology.

Weaknesses:

-The revised version has addressed previous concerns.

Likely Impact:

- The ion channel and synapse-related gene cassettes that distinguish the primary motor neuron circuit are shared with some mammalian circuits that also generate fast, reliable synaptic transmission.

- The transcriptomic data have been deposited in the publicly accessible Gene Expression Omnibus allowing others to mine the rich data set that also included glial cells that were not the focus of this study.

---

## [Referee Report · Reviewer #3 (Public Review)]

Functional and anatomical studies of spinal circuitry in vertebrates have formed the basis of our understanding of neuronal control of movements. Larval zebrafish provide a simplified system for deciphering spinal circuitry. In this manuscript, the authors performed scRNAseq on spinal cord neurons in larval zebrafish, identifying major classes of neuronal and glial types. Through transcriptome analysis, they validated several key interneuron types previously implicated in zebrafish locomotion circuitry. The authors went beyond identifying transcriptional markers and explored synaptic molecules associated with the strength of motor output. They discovered molecular distinctions causally related to the unique physiology of primary motoneuron (PMn) function, which involves providing strong synaptic outputs for escapes and fast swimming. They defined functional 'cassettes' comprising specific combinations of voltage-dependent ion channel types and synaptic proteins, likely responsible for generating maximal motor outputs.

Comments on revised version:

"However, the reviewer interprets Figure 2c to show that Type I, not Type II, V2a is more highly recruited over the range of higher swimming speeds whereas we conclude just the opposite."

BRE: The preceding is the authors' response to the Reviewer's critique of Version 1 of the manuscript and refers to Figure 2C of Menelaou and McLean, Nat Commun. 10:4197, 2019; PMID: 31519892; PMCID: PMC6744451. Below the Reviewer's second critique elaborates on this point. The authors chose not to modify the manuscript further.

This is not what I would like to maintain in my previous report. Both Type I and Type II V2a neurons are recruited during very fast swimming (70 Hz). The degree of the de-recruitment of Type I V2a neurons during slower swimming (40-60 Hz) is larger than Type II. Thus, what I would like to say is that Type I V2a neurons are more analogous to PMns than Type II V2a neurons (Both PMns and SMns are recruited during very fast swimming, and PMns tend to be de-recruited during slower swimming).

In this sense, I don't like the author's way of relating Type II V2a neurons to escapes and very fast swimming. However, if the authors insist on the current form of the manuscript, I do not strongly object.

---

## [Author Response]

The following is the authors’ response to the original reviews.

**Reviewer 1**

We now make clear throughout the manuscript that our proposition, holding the fast cassette as central to control over powerful movements governed by the PMn, remains a hypothesis. However, we provide additional rationale for our thinking that this is the case based on functional distinctions between the PMns and SMns. Both reviewers 1 and 2 also questioned why so few synaptic and ion channel genes are seen for the SMn type. As pointed out by the reviewer, the idea that small differences in birthdates between Mn types seems like an unlikely explanation and was removed. Now, we better develop the idea that the low levels of expression of both ion channel and synaptic genes in SMns are consistent with the finding from electrophysiology that point to greatly lowered levels of transmitter release, compared to PMns. Additionally, for the purpose of identifying all synaptic and ion channel genes shared equally between Mn types, we re-examined the transcriptome. Figure 7A & B now reflect all genes in these two categories detected above threshold in PMn and SMn types, and not just examples.

**Reviewer 2**

We have added cell types in mammalian circuits shown to express the ion channel cassette members. Examples include the calyx of Held in the auditory circuit and the cerebellar Purkinje neurons. As we show with zebrafish PMn these mammalian neurons form fast, reliable circuits. In these cases, it is noteworthy that our proposal is the first to link all three as functional partners in fast AP firing and high-fidelity synaptic transmission. The suggestion that pancreatic cells would be represented in our data is deemed highly unlikely as our technique separated out the spinal cords prior to dissociation. Finally, as suggested, we added the disclaimer that we can not exclude the possibility that clusters sharing both glia and neuronal markers may represent cell doublets. Other minor corrections were all made.

**Reviewer 3**

First, we agree that the role of PMns is not restricted to escape behavior. They have been shown to participate in the highest speed of swimming as well. We have made this clear throughout the paper.

Second, we are at odds with this reviewer over the Type I and Type II V2a recruitment during high speed swimming. We agree that both V2a types of interneurons are involved in high speed swimming and likely escape, as both directly innervate the PMns, as pointed out by the reviewer in Figure 2c of Menelaou and McLean 2019. However, the reviewer interprets Figure 2c to show that Type I, not Type II, V2a is more highly recruited over the range of higher swimming speeds whereas we conclude just the opposite. These data, along with other papers we cited, have been firmed up in the text to support a central role played by Type II.

Third, the reviewer recommends we remove Figures 6b and 6c relating to our two newly discovered SMn markers, fox1b and alcamb. Our data shown in Figure 6a shows that these markers label SMn somas in two distinct layers along the dorsal-ventral axis in the spinal cord. The reviewer objects to Figures 6b and 6c which compare the location of our two markers to the distributions of two well studied SMn labeling transgenic lines, islet:GFP and gata2:GFP. The correspondence is not absolute but suggests that the fox1b labels islet SMns and alcamb labels the gata2 SMns. In the previous version of the paper, we suggested that this correspondence might further signal different dorsal-ventral projections. This suggestion was based solely on reports that islet and gata2 transgenic lines preferentially label SMns with different projections. We do not view this particular point as important and in light of the controversy surrounding these projections, as noted by the reviewer, we removed all reference to the subject of muscle target areas. We focus instead, on our finding of two new markers that label different dorsal ventral soma layers which MAY correspond to previously described SMn types. This reasoning is made clear in the manuscript and, because of its potential importance, we elected to retain Figures 6b and 6c as a call for future testing.

The reviewer makes other suggestions that were all incorporated. The CoLo estimates indeed were too high, as questioned by the reviewer, because, early on, we inadvertently counted two clusters rather than the single cluster that was later authenticated. This has been corrected to reflect 1.1% in Table 1. The evx1 and evx2 data have been added to Figure 4C. Nomenclature is corrected for KA neurons. We make clear that the axonal projections for CoLo were made with mCherry expression not the in-situ label. The Hayashi reference was added.